



# Estimating Snow Depth on Arctic Sea Ice using Satellite Microwave Radiometry and a Neural Network

Anne Braakmann-Folgmann[1] and Craig Donlon[1]

[1]European Space Agency, Keplerlaan 1, 2201AZ Noordwijk, The Netherlands

**Correspondence:** Anne Braakmann-Folgmann (anne.bf@gmx.de)

**Abstract.** Snow lying on top of sea ice plays an important role in the radiation budget because of its high albedo, the Arctic freshwater budget, and influences the Arctic climate: it is fundamental climate variable. Importantly, accurate snow depth products are required to convert satellite altimeter measurements of ice freeboard to sea ice thickness (SIT). Due to the harsh environment and challenging accessibility, in situ measurements of snow depth are sparse. The quasi-synoptic frequent repeat
coverage provided by satellite measurements offers the best approach to regularly monitor snow depth on sea ice. A number of algorithms are based on satellite microwave radiometry measurements and simple empirical relationships. Reducing their uncertainty remains a major challenge.

A High Priority Candidate Mission called the Copernicus Imaging Microwave Radiometer (CIMR) is now being studied at the European Space Agency. CIMR proposes a conically scanning radiometer having a swath > 1900 km and including channels
at 1.4, 6.9, 10.65, 18.7 and 36.5 GHz on the same platform. It will fly in a high inclination dawn-dusk orbit coordinated with the MetOp-SG(B). As part of the preparation for the CIMR mission, we explore a new approach to retrieve snow depth on sea ice from multi-frequency satellite microwave radiometer measurements using a neural network approach. Neural networks have proven to reach high accuracies in other domains and excel in handling complex, non-linear relationships. We propose one neural network that only relies on AMSR2 channel brightness temperature data input and another one using both AMSR2 and
SMOS data as input. We evaluate our results from the neural network approach using airborne snow depth measurements from Operation IceBridge (OIB) campaigns and compare them to products from three other established snow depth algorithms. We show that both our neural networks outperform the other algorithms in terms of accuracy, when compared to the OIB data and we demonstrate that plausible results are obtained even outside the algorithm training period and area. We then convert CryoSat freeboard measurements to SIT using different snow products including the snow depth from our networks. We confirm that a
more accurate snow depth product derived using our neural networks leads to more accurate estimates of SIT, when compared to the SIT measured by a laser altimeter at the OIB campaign. Our network with additional SMOS input yields even higher accuracies, but has the disadvantage of a larger "hole at the pole". Our neural network approaches are applicable over the whole Arctic, capturing first-year ice and multi-year ice conditions throughout winter. Once the networks are designed and trained, they are fast and easy to use. The combined AMSR2 + SMOS neural network is particularly important as a pre-cursor
demonstration for the Copernicus CIMR candidate mission highlighting the benefit of CIMR.



# 1 Introduction

Climate change and globalisation are the dominant drivers of societal impacts in the Arctic with economic development rapidly transforming the geo-politics and the physical and biogeochemical environment of the region. For example, new prospectors are increasing their activities using modern techniques for oil and gas, fisheries and mineral resources, and commercial ship traffic is growing dramatically. In this context, snow depth is an important parameter for climate studies, modelling and forecasting. Snow on sea ice strongly influences the Earth's radiation budget with its high albedo and acts as an insulation controlling sea ice growth and melt. In the melt season snow on sea ice contributes to the freshwater input and inhibits deep ocean circulation because of surface freshwater stratification. Additionally, to retrieve sea ice thickness (SIT) from laser (NASA ICESat) or radar altimeter (e.g European Space Agency (ESA) CryoSat) freeboard measurements, snow depth has to be known with a high accuracy. The uncertainty in today's snow on sea ice products contributes significantly to the uncertainty in SIT (Zygmuntowska et al. (2014), Giles et al. (2007)). Ship traffic across the northern Sea Route in the Arctic is increasing and will further increase as sea ice retreats. To navigate through the sea ice, SIT is a key parameter, but also the snow depth itself is relevant due to its very high friction (Huang et al. (2018)).

To derive SIT from CryoSat freeboard the Warren climatology product (Warren et al. (1999)) is often used. It relies on snow depth measurements collected from manned drifting stations and isolated locations (reached via aircraft) over multi-year ice (MYI) in the Arctic between 1954 and 1991. These measurements are summarised in monthly maps and contour lines of snow depth have been derived. For lack of a better operational product, this climatology is still widely used - sometimes with a modification factor of 0.5 or 0.7 to account for lower snow depths on first-year ice (FYI) and the fact that less ice survives each summer (Kwok and Cunningham (2015)). Obvious drawbacks of this climatology are that it is outdated (Kern et al. (2015)), that it was collected mostly over MYI, its quite broad spatial resolution and that it does not allow for any interannual variation.

The quasi-synoptic frequent repeat coverage provided by satellite measurements offer an excellent approach to regularly monitor snow depth on sea ice. Satellite microwave measurements offer a clear advantage over visible or thermal infrared techniques because they penetrate through clouds and deliver measurements during the long polar night. Unfortunately, at this time the frequencies of primary interest (1.4 – 7.0 GHz) are characterized by a large surface footprint. Measurements made at higher frequency (18-89 GHz) are used to derive estimates of snow depth on sea ice with varying degrees of success. A number of algorithms are based on simple empirical relationships to in situ measurements and reducing the uncertainty in derived snow depth products remains a major challenge. The first algorithm that was developed using satellite microwave radiometer data is reported in Markus and Cavalieri (1998). It uses an empirical relation between the gradient ratio of the 37.0 GHz and 19.4 GHz channels of the Special Sensor Microwae/Imager (SSM/I) sensor together with in situ and ship observations of snow depth in Antarctica. Comiso et al. (2003) modified the Markus and Cavalieri (1998) algorithm coefficients to match the slightly different frequencies of the Advanced Microwave Scanning Radiometer (AMSR-E) and follow on AMSR2 mission. This algorithm only produces reasonable results over FYI and in general the use of microwave radiometer data is limited to cold and dry snow conditions, because in the melt season wet snow acts as a black body (Markus et al. (2006)).



Recently it was argued that the use of lower frequencies (e.g. 6.9 GHz) that measure microwave emissions deeper in the snow layer could improve the accuracy and allow the retrieval of larger snow depths, since the 36.5 GHz signal is saturated at around 50 cm (Markus et al. (2006)). Rostosky et al. (2018) proposed such an algorithm using the gradient ratio between 6.9 GHz and 18.7 GHz. Furthermore their algorithm enables an extension to MYI by using two separate empirical fits for FYI
and MYI.

More recently, Kilic et al. (2018b) make use of the low frequency 6.9 GHz channel, but instead of using the gradient ratio, they fit a multilinear regression between microwave radiometer data and snow depth using data from Ice Mass Balance (IMB) buoys in the Arctic.

In general microwave radiometer observations are widely used input data for snow depth retrieval. They benefit from a
long data record, allow at least daily coverage over the poles and most importantly are independent of weather and darkness. The only drawback is the rather broad spatial resolution (AMSR2 has a 35 x 62 km footprint at 6.9 GHz). The European Space Agency (ESA) is now studying a High Priority Candidate Mission (HPCM) called the Copernicus Imaging Microwave Radiometer (CIMR, Donlon et al. (2019)). CIMR proposes a conically scanning radiometer having a swath > 1900 km and will include channels at 1.4 GHz (60 km), 6.9 and 10.65 GHz (<15 km), 18.7 GHz (5-6 km) and 36.5 GHz (4-5 km) on the same
platform. The mission will occupy a high inclination dawn-dusk orbit coordinated with the MetOp-SG(B) satellite offering opportunities for synergy with the Microwave Imager (MWI) and Scatterometer (SCA). CIMR would not only guarantee continuity in microwave radiometer observations, but it would also ensure continuity at low frequency L-band (1.4 GHz), currently provided by ESA's Soil Moisture and Ocean Salinity (SMOS) and NASA's Soil Moisture Active Passive (SMAP) satellite and for the first time provide L-band and higher frequency measurements on the same platform in a high inclination
orbit.

Maaß et al. (2013) demonstrate the possibility to determine snow depth from 1.4 GHz brightness temperatures measured by SMOS. The insulation of the snow cover leads to increasing brightness temperatures at 1.4 GHz correlated with snow depth. Maaß et al. find that the effect is more pronounced at horizontal polarization. The approach works well for thick sea ice (ice thicker than 1-1.5 m) and snow depths of 35 cm. Also Zhou et al. (2018) developed a combined snow depth and SIT retrieval
approach from a combination of SMOS data with laser altimetry incorporating a radiation model.

Yet another possibility to determine snow depth is to exploit the different scattering horizons from CryoSat (Ku-band) and SARAL/AltiKa (Ka-band) (Guerreiro et al. (2016), Lawrence et al. (2018)). The same concept may be applied to the upcoming overlap of CryoSat and ICESat-2 (Lawrence et al. (2018)). ESA currently also investigates the Copernicus polaR Ice and Snow Topography ALtimeter (CRISTAL, Kern et al. (2019)) as a High Priority Candidate Mission. If selected, CRISTAL
would uniquely offer co-temporal Ku- and Ka-band measurements in a high inclination orbit. In comparison to microwave radiometer measurements, however, the temporal coverage would be quite low due to the small nadir-only footprint of the altimeter although repeat global sampling every 10 days is anticipated.

The opportunity for synergy and inter-calibration between multi-frequency altimetry (e.g. CRISTAL) and CIMR snow depth retrievals over sea ice is obvious. As part of the preparation for the future CIMR mission, we explore a new approach to retrieve
snow depth on sea ice from satellite microwave radiometer measurements using a neural network approach. Neural networks



provide a technique to model any complex, non-linear relationship, including the multi-frequency microwave signal emissions from within a snow layer. The application of neural networks for this purpose is still developing but a few simple attempts exist: Tedesco et al. (2004) apply a simple neural network with one hidden layer to derive snow depth and SWE on land. They use the 19 and 37 GHz brightness temperatures at both polarizations as input.

We build a deeper, more advanced neural network to retrieve snow depth on sea ice from satellite microwave radiometer measurements and train our network with Operation Ice Bridge (OIB) snow depths (Kurtz et al. (2013)) in the Arctic. We build on the algorithms by Markus and Cavalieri and Rostosky et al. using both the 'traditional' 36.5/18.7 gradient ratio and the lower frequency 18.7/6.9 gradient ratio as input together with polarization ratios. We also explore the use of SMOS together with AMSR2 data as input for one of our neural networks. Our neural networks are applicable over both FYI and MYI ice and

no additional ice type product is needed to differentiate between both. Once designed and trained, they are fast and easy to use and would also work with future measurements from the CIMR radiometer.

    We verify our neural network approaches with another part of the OIB data and compare the results to the snow depth algorithms by Markus and Cavalieri , Rostosky et al. and Kilic et al. . We also evaluate how the different snow products influence the SIT retrieval from CryoSat freeboard data.

In the next section we summarise the different snow depth algorithms used for comparison, introduce our neural network approach and explain the SIT calculation. In section 3 we introduce the data used for training, evaluation and the comparisons. The results are then shown and discussed in section 4 before we end with a conclusion.

## 2    Methodology

First we review a few existing algorithms for snow depth on sea ice calculation from satellite microwave radiometer brightness

temperatures, before we introduce our own neural network approach. The neural network somehow builds upon the findings of these more traditional algorithms and will also be compared to them in section 4.

### 2.1    Snow depth from Markus and Cavalieri

Markus and Cavalieri (1998) developed the first algorithm to retrieve snow depth $h_s$ on sea ice from passive microwave measurements in 1998. This algorithm is still widely used and is based on the gradient ratio between 19 GHz and 37 GHz

brightness temperatures Tb at vertical polarization $V$:

$$h_s[\text{cm}] = 2.9 - 782 \cdot \frac{\text{Tb}_{\text{ice}}(37V) - \text{Tb}_{\text{ice}}(19V)}{\text{Tb}_{\text{ice}}(37V) + \text{Tb}_{\text{ice}}(19V)} \tag{1}$$



$Tb_{ice}$ is the brightness temperature of the ice covered part of the footprint. This correction is important, since we are only interested in the change of brightness temperature due to snow cover and otherwise the open water part would dominate the signal. It is calculated from:

$$Tb_{ice}(f,p) = \frac{Tb(f,p) - (1 - SIC) \cdot Tb_{OW}(f,p)}{SIC} \tag{2}$$

$Tb_{OW}(f,p)$ is the open water tie point for frequency $f$ and polarization $p$ and SIC is sea ice concentration. In the equations we round the frequency to the nearest integer and indicate vertical linear polarisation with a $V$ and horizontal linear polarisation with an $H$. Originally the two linear regression coefficients were derived from a fit of SSM/I brightness temperatures to Antarctic in situ and ship observations. Comiso et al. (2003) updated the algorithm coefficients to fit the slightly different incidence angle and frequencies of AMSR-E. The same coefficients are also applied for the Arctic.

The algorithm is limited to dry, cold snow, which is thinner than 50 cm and should only be applied over FYI (Markus et al. (2006)). Instead of the original values, we use the coefficients from Comiso et al. as given in Eq. 1, open water tie point values for AMSR2 from Ivanova et al. (2014) and calculate the SIC with the Nasa Team algorithm (Cavalieri et al. (1984)). To be comparable with the other algorithms, we ignore the shortcomings of the algorithm over MYI and apply it Arctic wide anyway. This is also an essential requirement when applied in SIT retrieval.

## 2.2   Snow depth from Rostosky et al.

Rostosky et al. (2018) follow a similar approach as Markus and Cavalieri using a gradient ratio and two linear regression coefficients. However, instead of using the gradient ratio between 18.7 GHz and 36.5 GHz, they apply the 6.9 GHz to 18.7 GHz gradient ratio. The lower frequencies enable a determination of snow depths exceeding 50 cm (due to microwave emissions emanating from deeper within the snow at this frequency), where the 36.5 GHz channel becomes saturated. Furthermore a

simulation by Markus et al. (2006) and a correlation analysis by Rostosky et al. (2018) suggest a stronger relation of snow depth to this gradient ratio. To use this gradient ratio, they determined a new set of regression coefficients by fitting AMSR-E and AMSR2 brightness temperatures to OIB snow depth. They exclude single years for verification and validation work. Furthermore they extend the approach to be applicable over both FYI and MYI, while Markus and Cavalieri 's approach was found to deliver reasonable results only over FYI. The extension to MYI is achieved by fitting a second set of parameters to

the MYI covered part of the OIB data. We use the coefficients determined with OIB data from 2009 - 2014, since our test data is from 2015. When applying this algorithm, the ice type (FYI or MYI) must be known with confidence. They use the ice type product from OSISAF - derived by a combination of microwave radiometry and scatterometer data (Aaboe et al. (2016)) and discard areas where the ice type is not known with high confidence (confidence level < 4). On FYI snow depth is calculated from

$$h_s[cm] = 19.74 - 556.69 \cdot \frac{Tb_{ice}(19V) - Tb_{ice}(7V)}{Tb_{ice}(19V) + Tb_{ice}(7V)} \tag{3}$$



and on MYI from

$$h_s[cm] = 18.73 - 376.32 \cdot \frac{\mathrm{Tb}_{\mathrm{ice}}(19V) - \mathrm{Tb}_{\mathrm{ice}}(7V)}{\mathrm{Tb}_{\mathrm{ice}}(19V) + \mathrm{Tb}_{\mathrm{ice}}(7V)}. \tag{4}$$

Again, SIC from the NASA Team algorithm is used to correct for the open water part within the footprint (Eq. 2).

## 2.3 Snow depth from Kilic et al.

Kilic et al. (2018b) developed a simple multilinear regression approach using vertically polarised brightness temperatures at 6.9 GHz, 18.7 GHz and 36.5 GHz. These three channels were identified as the best predictor combination in a forward selection method with the OIB data of 2013. They then derived the multilinear regression coefficients from a fit of AMSR2 brightness temperatures with the data from four IMB buoys (2012G, 2012H, 2012J and 2012L) to yield the following formula:

$$h_s[cm] = 177.01 + 1.75 \cdot \mathrm{Tb}(7V) - 2.80 \cdot \mathrm{Tb}(19V) + 0.41 \cdot \mathrm{Tb}(37V) \tag{5}$$

They use SIC charts from the European Centre for Medium-Range Weather Forecasts (ECMWF) Re-Analysis Interim (ERA-Interim) data and discard areas outside 100% SIC. To be consistent with the other approaches we use the OSISAF SIC product (Lavergne et al. (2019)) and discard areas with SIC lower than 80%.

## 2.4 Snow depth from our Neural Network Approach

Artificial neural networks are a means of machine learning inspired by the human brain to learn higher-order representations and perform diverse tasks. In contrast to other machine learning techniques they are designed to extract relevant features and their weighting in the model themselves. Deep neural networks allow to learn higher-order representations, to tackle more complex problems and outperform other means of machine learning in terms of accuracy (Schmidhuber (2015)). Neural networks can be viewed as a universal system to represent any function. Instead of designing representative features or building a complex physical model, the challenge with neural networks is to design an appropriate architecture.

We design our neural networks with the framework Keras (Chollet et al. (2015)), using Tensorflow (Abadi et al. (2015)) backend. Three inputs from AMSR2 are used in our neural networks: the gradient ratio between vertically polarised brightness temperatures at 18.7 GHz and 36.5 GHz, as proposed by Markus and Cavalieri , the gradient ratio between vertically polarised brightness temperatures at 6.9 GHz and 18.7 GHz, as used by Rostosky et al. , and the polarization ratio PR between vertically and horizontally polarised brightness temperatures at 36.5 GHz:

$$\mathrm{PR}(37) = \frac{\mathrm{Tb}_{\mathrm{ice}}(37V) - \mathrm{Tb}_{\mathrm{ice}}(37H)}{\mathrm{Tb}_{\mathrm{ice}}(37V) + \mathrm{Tb}_{\mathrm{ice}}(37H)} \tag{6}$$

This polarization ratio is also used to differentiate between FYI and MYI by Comiso (2012), so it seems likely that this information is not directly correlated with snow depth, but rather with the ice type and that the neural network uses this input in a similar manner as the approach by Rostosky et al. which requires independent ice type information. We also experimented with other combinations, gradient- and polarization ratios, as well as using the brightness temperatures directly as input. This

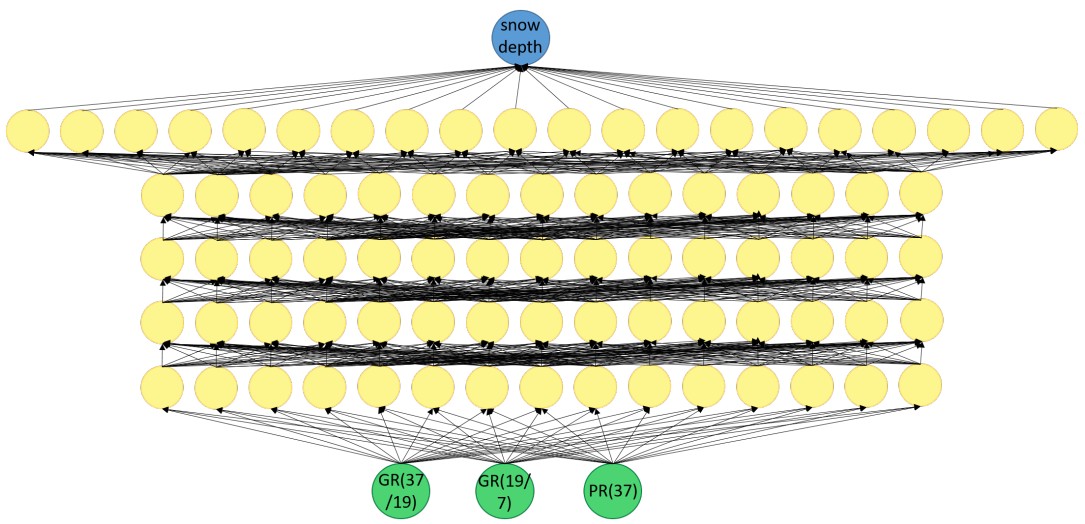

**Figure 1.** Architecture of the AMSR2-only neural network: the 37/19 gradient ratio, the 19/7 gradient ratio and the polarization ratio at 37 GHz are used as input (green circles). They are transformed by five fully connected hidden layers with fifteen or twenty neurons each (yellow circles) to finally produce snow depth as output (blue circle). Each neuron (circle) has a bias and each connection (arrow) is associated with a different weight.

choice yields the best results. Just as Markus and Cavalieri , we also apply a correction for the open water part within the footprint, using $\mathrm{Tb_{ice}}$ with SIC from the NASA Team algorithm in our gradient and polarization ratios (Eq. 2).

The first (AMSR2-only) neural network consists of five fully connected hidden layers with 15 neurons in each of the first four hidden layers and 20 neurons in the last hidden layer (see Figure 1 for an illustration). The number of layers and neurons

was empirically found to work best for this specific set up. A few rules of thumb exist for the design of a good neural network architecture, but to a large part it is subject to trying different set-ups and observing the error on the validation data set.

In a fully connected neural network all neurons of the previous layer are connected to each neuron of the next layer and each connection is associated with a different weight. Furthermore each neuron can have a different bias. The output from each neuron in the first hidden layer is given by the sum of all three weights (the arrows connected to this neuron in figure 1) times

10 the three inputs plus the bias of this neuron. To introduce a non-linearity and enable the network to learn non-linear relations, the output of each layer may be transformed by a so called activation function $\phi$. Taking into account all fifteen outputs from the first hidden layer $\boldsymbol{h}$ we can write this as a matrix vector multiplication, where $\mathbf{W}$ is a 15x3 weight matrix, $\boldsymbol{x}$ the 3x1 input vector and $\boldsymbol{b}$ a 15x1 bias vector:

$$\boldsymbol{h} = \phi(\mathbf{W} \cdot \boldsymbol{x} + \boldsymbol{b}). \tag{7}$$

The hidden layers store the features or the information extracted from the input. The different weights and biases allow each neuron to focus on a different aspect. Usually the features get more abstract and complex the deeper the network becomes,



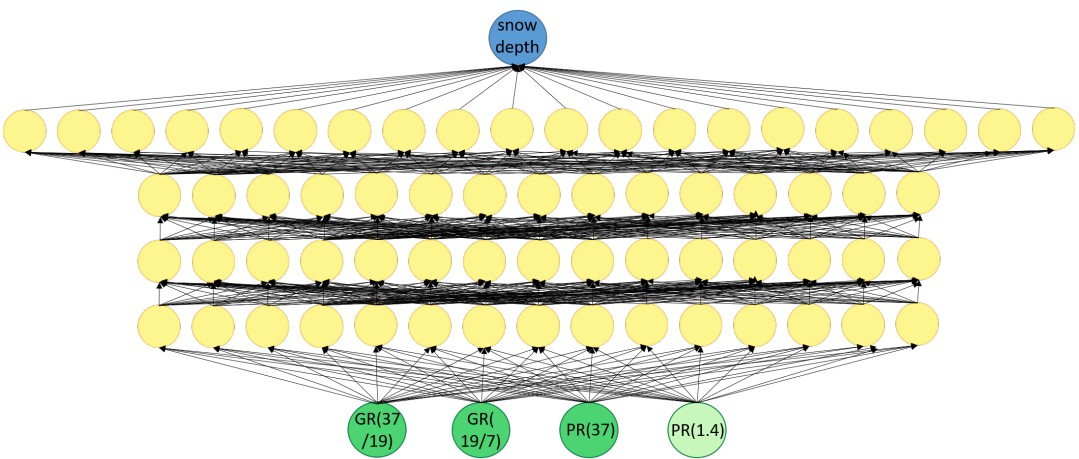

**Figure 2.** Architecture of the AMSR2 + SMOS neural network: the 37/19 gradient ratio, the 19/7 gradient ratio and the polarization ratio at 37 GHz are used as input from AMSR2 (green circles) and the polarization ratio at 1.4 GHz is used as input from SMOS (light green circle). They are transformed by four fully connected hidden layers with fifteen or twenty neurons each (yellow circles) to finally produce snow depth as output (blue circle).

since each subsequent layer is created from the already transformed features of the previous layer. The output layer consists of one neuron and represents the estimated snow depth.

To allow the network to learn non-linear relationships, as we expect them to occur in the emission and scattering of a microwave signal in a snow layer, we apply activation functions $\phi$. After the first hidden layer we apply a sigmoid function, after

all subsequent hidden layers we apply the rectified linear unit (ReLU) activation function and finally the output is transformed by a hyperbolic tangent activation function.

Batch normalization (Ioeffe and Szegedy (2015)) makes a neural network less sensitive to the random initialization of the weights and biases and improves its generalization capabilities. The use of batch normalization is recommended for deep neural networks and with sigmoidal non-linearities. Therefore we include batch normalization after the first hidden layer.

Training a neural network means to slightly change its weights and biases step by step to minimise a loss function. This technique is known as Stochastic Gradient Descent. The Adam optimiser (Kingma and Ba (2015)) is a more elaborate extension of Stochastic Gradient Descent, which we use to train our network in 250 epochs using a batch size of 30. We choose the mean absolute percentage error between the estimated snow depth and the OIB snow depth as our loss function.

The design of the second neural network combining AMSR2 and SMOS input is very similar to the first one. In addition to

the three AMSR2 inputs, we add the polarization ratio between vertically and horizontally polarised brightness temperatures at 1.4 GHz from SMOS as a fourth input node. This is calculated analogously to Eq. 6. At 1.4 GHz, SMOS provides a means to penetrate deeper into the snow layer. We also tested gradient ratios between the AMSR2 and SMOS channels and using brightness temperatures at 1.4GHz directly. The polarization ratio gave the best results. This is the only time we do not account for open water within the footprint and we do not use open water tie points to correct the SMOS brightness temperatures. We



calculated some open water tie points at 1.4 GHz and applied Eq. 2, but this slightly degraded the network's performance, so we choose not use them. Because SMOS coverage has a large hole at the pole, we have less data for training, validation and testing. Therefore we also reduce the number of parameters of the neural network and delete one hidden layer with 15 neurons. Otherwise the network is identical to the AMSR2-only network. Figure 2 illustrates the design of the combined AMSR2 +
SMOS network.

## 2.5  Sea ice thickness

Sea Ice Thickness (SIT) can be calculated from the sea ice freeboard $h_{\mathrm{fb}}$ measured by CryoSat assuming hydrostatic equilibrium:

$$\mathrm{SIT} = \frac{\rho_w \cdot h_{\mathrm{fb}} + \rho_s \cdot h_s}{\rho_w - \rho_i}. \tag{8}$$

$\rho_w$ is the density of sea water, which we set to 1025 $\frac{kg}{m^3}$ (Alexandrov et al. (2010)), $\rho_i$ the ice density and $\rho_s$ the snow density. For the snow density we assume a bulk value of 320 $\frac{kg}{m^3}$, as suggested by the Warren climatology (Warren et al. (1999)) of March and April (when the OIB data was collected). The ice density depends on the age of the sea ice and Alexandrov et al. (2010) found a mean value of 882 $\frac{kg}{m^3}$ for MYI and 917 $\frac{kg}{m^3}$ for FYI. Both values can be weighted according to the MYI fraction as suggested by Kwok and Cunningham (2015). However, King et al. (2018) found that using only the MYI value of
882 $\frac{kg}{m^3}$ agrees better with helicopter-borne electromagnetic SIT sounding measurements. We observe the same in comparison to the OIB SIT measurements, and therefore apply 882 $\frac{kg}{m^3}$ everywhere.

The last - and a major - uncertainty in the calculation of SIT is snow depth $h_s$ (Zygmuntowska et al. (2014), Giles et al. (2007)). Here we use the Warren climatology and the algorithms from Markus and Cavalieri , Rostosky et al. , Kilic et al. , our neural networks and also the snow depth measured directly by the OIB snow radar to see how different snow products
influence SIT.

## 3  Data

### 3.1  Operation Ice Bridge (OIB)

Operation Ice Bridge (OIB) was a flight campaign conducted in March and April 2009–2015 by NASA (Kurtz et al. (2013)). The onboard snow radar provides snow depth measurements by identifying both the air–snow and snow–ice interface within
the radar returns. This time difference can then be converted to snow depth, if the snow density is known. Furthermore, a combination of the onboard laser altimeter (tracking the ice + snow freeboard) with snow depth allows the calculation of SIT (Farrell et al. (2012)). We use OIB data from the Round Robin Data Package (RRDP) Version 2. This dataset was developed as part of ESA's sea ice climate change initiative (CCI) project and can be downloaded from http://www.seaice.dk/RRDB-v2/. It contains the OIB snow depth and SIT data together with collocated AMSR-E or AMSR2 data. In this study we only use data

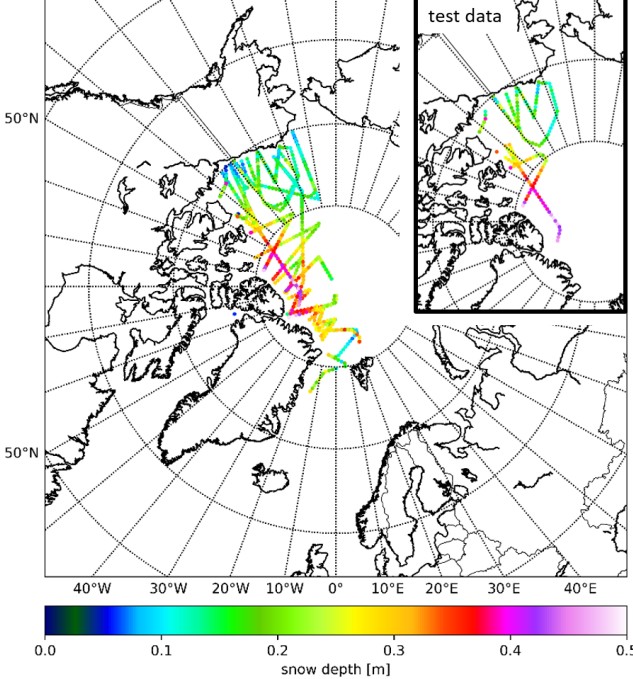

**Figure 3.** Flight tracks of the Operation Ice Bridge (OIB) campaigns 2013, 2014 and 2015. The measured snow depth is colour-coded. The box on the top right shows which part of the OIB data is used as test data.

from the years 2013–2015, where AMSR2 data is available, to avoid the need for an AMSR-E versus AMSR2 inter-calibration. The OIB data in this RRDP stem from NSIDC and OIB snow depth data are averaged into 50 km sections for a better overlap and collocation with AMSR (Pedersen and Saldo (2018)).

The OIB measured snow depth and SIT were compared to ground-based in situ measurements along a 2 km transect from
the Danish GreenArc sea ice camp across different ice types. Both snow depth and SIT were found to agree very well with in situ data (mean difference 0.01 m and 0.05 m respectively, (Farrell et al. (2012)). Also a comparison to in situ snow depth measurements from the Bromine, Ozone, and Mercury Experiment (BROMEX, Webster et al. (2014)) and to a reconstruction of snow depth from snowfall reanalysis data and sea ice motion (Blanchard-Wrigglesworth et al. (2018)) show good agreement. Therefore we regard the OIB data as the best available validation data set and use a part of it to train our neural network and
another part for evaluation. To train the network we temporally divide the OIB data into a training- (70%), a validation- (15%) and a test (15%) dataset. This gives us 755 valid snow depth measurements in 2013 and 2014 for training, 162 valid measurements in 2014 and 2015 for validation – meaning the identification of the best network architecture – and 162 valid snow depth measurements in 2015 for testing. Figure 3 shows the flight tracks and the measured snow depths from the 2013– 2015 campaigns. The top right box illustrates which parts are used as test data. When we train the AMSR2 + SMOS neural
network, we have to discard all areas (especially the bigger hole at the pole), where no SMOS data is available. Again we split




the remaining data into 70% for training and 15% for validation and testing each. We then end up with only 299 valid data points for training, 64 for validation and 65 for testing.

## 3.2 AMSR2

To train the neural networks and for all comparisons with OIB data, we use the collocated AMSR2 brightness temperatures
provided in the RRDP. For all other purposes (longer time series and maps of the whole Arctic), we use the AMSR2 L1R brightness temperature swath data from JAXA (available at ftp.gportal.jaxa.jp). In the L1R product all frequencies are resampled to the 6.9 GHz resolution and centred at the center of the 89 GHz A footprint (Maeda et al. (2016)). Since CIMR would provide the same frequencies that we are using and a similar L1R product, our neural network could directly be applied to CIMR data and would provide snow depth at a higher spatial resolution.

## 3.3 SMOS

We use daily L3 SMOS data from the Centre Aval de Traitement des Données SMOS (CATDS), available at https://www.catds.fr /sipad/. This product is derived from L1C by gridding it to the 25 km global EASE-2 grid. RFI filtering is applied and certain SMOS L1C flags are taken into consideration. The brightness temperatures are available at full linear vertical and horizontal polarization and averaged into 5° incidence angle bins (Al Bitar et al. (2017), Kerr et al. (2013)). We average the ascending
and descending tracks and two of those incidence angle bins to receive brightness temperatures around 55° (50–60°) incidence angle, as CIMR would measure them (Kilic et al. (2018a)). To collocate the SMOS data with the OIB and AMSR2 data from the RRDP, we average SMOS measurements within 25 km from the OIB position of the same date.

## 3.4 CryoSat

For the calculation of SIT, we use the freeboard data in the Geophysical Data Record (GDR) product from the CryoSat-2
Science server: http://science-pds.CryoSat.esa.int. Flagged freeboard data are excluded. To compare the CryoSat-derived SIT with the OIB SIT, we need to collocate the CryoSat freeboard with the OIB measurements. For each OIB SIT measurement we average all CryoSat measurements within 25 km from the OIB position and within +/- 10 days to the OIB flight, assuming that SIT does not change so quickly. Doing so, we take the mean of on average 296 CryoSat freeboard measurements (median: 167 CryoSat measurements) and thereby account for the much smaller footprint of CryoSat compared to the snow depth products
and the averaged OIB data in the RRDP, but also reduce the uncertainty of a single freeboard measurement.

## 3.5 Ancillary data

### 3.5.1 Ice concentration chart

All snow depth on sea ice algorithms, that are investigated here, rely on a SIC chart to apply them only in areas of at least 80% SIC. For this we apply the SIC product from OSISAF available at ftp://osisaf.met.no/reprocessed/ice/conc/v2p0 (Lavergne et
al. (2019)). Apart from Kilic et al. all algorithms also require SIC to correct the brightness temperatures for a potential open



water part within the footprint. For this purpose we apply the Nasa Team (Cavalieri et al. (1984)) algorithm, as suggested by Markus and Cavalieri . This is much faster than to map a gridded SIC chart to all swath data, but it also misidentifies a few areas in the open ocean as sea ice. To remove these we use the more accurate OSISAF SIC chart at the end.

### 3.5.2 Ice type product

The algorithm from Rostosky et al. requires reliable information on the ice type to distinguish FYI from MYI. As proposed in their paper, we also use the OSISAF ice type product (Aaboe et al. (2016)) from ftp://osisaf.met.no/archive/ice/type.

## 4 Results and discussion

### 4.1 Results on snow depth

In this section we measure the performance of our neural networks and compare the results to the algorithms proposed by Markus and Cavalieri (1998), Rostosky et al. (2018) and Kilic et al. (2018b). For this evaluation we employ the test data part of the OIB snow depth measurements. The performance is evaluated using the root mean squared error (RMSE), the correlation coefficient CC, the coefficient of determination ($R^2$) and the bias. These are defined as follows:

$$\text{RMSE} = \sqrt{\frac{\sum_{i=1}^{N}(f_i - y_i)^2}{N}} \tag{9}$$

$$\text{CC} = \frac{\sum_{i=1}^{N}(f_i - \bar{f})(y_i - \bar{y})}{\sqrt{\sum_{i=1}^{N}(f_i - \bar{f})^2 \cdot \sum_{i=1}^{N}(y_i - \bar{y})^2}} \tag{10}$$

$$R^2 = 1 - \frac{\sum_i (y_i - f_i)^2}{\sum_i (y_i - \bar{y})^2} \tag{11}$$

$$\text{bias} = \frac{\sum_{i=1}^{N}(f_i - y_i)}{N} \tag{12}$$

with $f_i$ being the estimated values from the algorithm, $y_i$ values from OIB and $\bar{y}$ or $\bar{f}$ the mean of the OIB or predicted values respectively.

Table 1 shows the results for the different algorithms. Here we use only those parts of the data, where AMSR2, SMOS and OIB data is available. This gives us 65 valid data points for testing. In terms of RMSE and the coefficient of determination the two neural networks (AMSR2-only NN and AMSR2 + SMOS NN) yield the best results, followed by the approach by Rostosky et al. . For Markus and Cavalieri one should keep in mind that we include snow depth estimates over MYI, where the algorithm is known to have issues. Concerning the correlation the algorithms by Rostosky et al. and Kilic et al. perform best,



|  | RMSE | CC | $R^2$ | bias |
|---|---|---|---|---|
| Markus & Cavalieri | 0.200 m | 0.75 | -4.37 | 0.182 m |
| Rostosky et al. | 0.075 m | **0.93** | 0.31 | 0.060 m |
| Kilic et al. | 0.132 m | **0.93** | -1.33 | 0.101 m |
| AMSR2-only NN | 0.053 m | 0.84 | 0.63 | **0.005 m** |
| **AMSR2 + SMOS NN** | **0.040 m** | 0.91 | **0.79** | -0.011 m |

**Table 1.** RMSE, correlation, coefficient of determination and bias between the different snow depth retrieval algorithms and OIB measured snow depth for all the test data where SMOS data is available

|  | RMSE | CC | $R^2$ | bias |
|---|---|---|---|---|
| Markus & Cavalieri | 0.194 m | 0.77 | -2.67 | 0.172 m |
| Rostosky et al. | 0.066 m | **0.90** | 0.58 | 0.040 m |
| Kilic et al. | 0.158 m | 0.89 | -1.44 | 0.120 m |
| AMSR2-only NN | 0.063 m | 0.82 | 0.61 | **0.003 m** |
| **AMSR2 + SMOS NN** | **0.059 m** | 0.85 | **0.66** | **-0.003 m** |

**Table 2.** RMSE, correlation, coefficient of determination and bias between the different snow depth retrieval algorithms and OIB measured snow depth for all the test data. When no SMOS data is available, the neural network with SMOS is equal to the neural network without SMOS

giving correlation coefficients of 0.93. Last but not least, the neural networks have essentially no bias (0.005 m and -0.011 m), while Rostosky et al. show the second smallest bias with 0.060 m. So overall both neural networks show very promising results and a higher agreement with OIB snow depth than the other algorithms. Comparing both neural networks with each other, we can easily conclude, that the addition of SMOS data further improves the neural network's accuracy - only the bias slightly increases.

To exploit also those parts of the OIB data, where no SMOS data is available, we now show the results on the full OIB test data set (162 valid data points for testing instead of 65). Figure 3 top right corner shows the whole test data set from OIB. It covers a range of snow depths on both FYI and MYI. The AMSR2 + SMOS neural network results stem from the AMSR2 + SMOS net, if SMOS data is available and from the AMSR2-only neural network otherwise. This ensures that we compare the same part of the data for all approaches and have more test data available. Combining the two networks could also be useful in a practical application to fill the hole at the pole and to still benefit from higher accuracies in regions, where SMOS is available.

Table 2 again shows the results for the different algorithms over the whole test data set. In terms of RMSE and the coefficient of determination the approach by Rostosky et al. and the neural networks again yield the best results (RMSE 0.066 m, 0.063 m and 0.059 m, $R^2$ 0.58, 0.61 and 0.66), with the AMSR2-only neural network being slightly better than Rostosky et al. and the combined neural network working best. Concerning the correlation also here the algorithms by Rostosky et al. and Kilic et al. outperform the others, giving correlation coefficients of 0.89 and 0.90. Last but not least, both neural networks have essentially





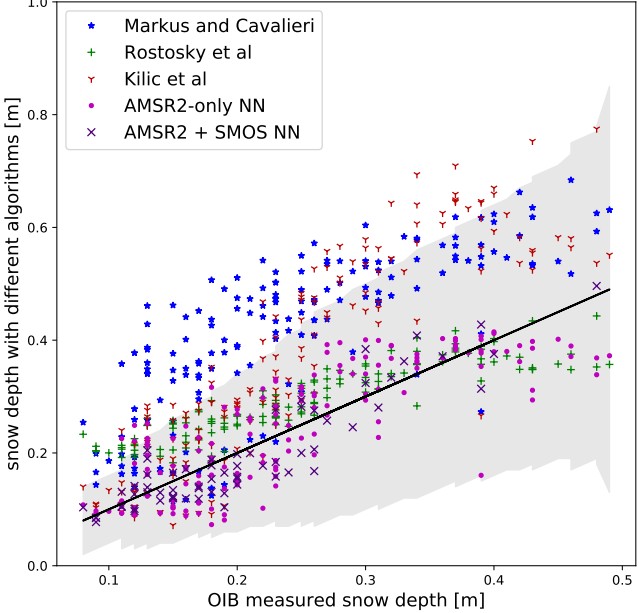

**Figure 4.** OIB measured snow depth (test data) versus predicted snow depth using different algorithms. Note that Markus and Cavalieri should only be applied over FYI, but this plot includes also MYI.

no bias (+/-0.003 m), while Rostosky et al. show the second smallest bias with 0.040 m. Overall we can conclude that the exact excerpt of the data does not make a big difference in terms of the conclusions and also here the neural networks perform best, with the combined neural network outperforming the AMSR2-only one. The difference between the two neural networks obviously is much larger and clearer, when only those parts of the data are used, where SMOS data is available. With CIMR

we expect to see the same significant improvement as demonstrated in Table 1 without the problem of losing data for training and testing.

For a visual impression we plot the estimated snow depth versus the snow depth measured by OIB in figure 4. The black line indicates a perfect match between the algorithm and OIB and the grey shaded region indicates the uncertainty range of the OIB snow depth measurements. In general the neural networks (pink dots for AMSR2-only and purple cross for AMSR2 + SMOS)

and the approach by Rostosky et al. (green plus) are closest to OIB. For Markus and Cavalieri (blue stars) we observe that low snow depths fit quite well, but larger snow depths are largely overestimated. We acknowledge that these high snow depths probably occur on MYI, where the algorithm is not well defined.

Most algorithms start to flatten at around 35-40 cm snow depth. This behavior can be explained by the saturation of the 36.5 GHz signal around this depth. The algorithm by Rostosky et al. is the only one, solely relying on lower frequency

channels and should not yet saturate at this depth. Indeed their estimation stays quite close to the OIB measurements, but also shows a slight decrease in slope. Anyway this might not be significant considering the small number of samples. For the





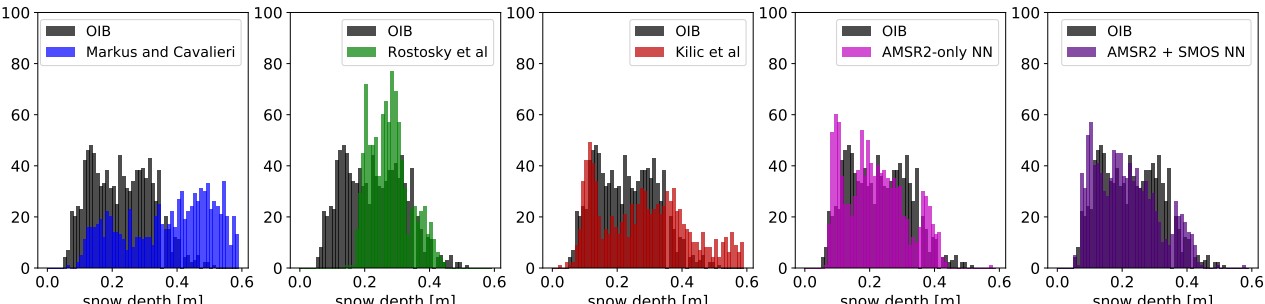

**Figure 5.** Distribution of OIB measured snow depth (all data) in grey versus estimated snow depth using different algorithms in colour.

AMSR2 + SMOS neural network, we do not observe a flattening, but we also only have very few samples available for high snow depth.

Figure 5 reveals the distribution of OIB snow depth in grey and the distribution of estimated snow depth in colour. To get a better idea of the algorithms' characteristics and to be statistically more meaningful, we show the results for the whole OIB dataset. For the test data the plots look similar, but less obvious. For Markus and Cavalieri (first plot in blue), we again observe that a lot of snow depths are highly overestimated - most likely due to the application of this algorithm over MYI (where it is poorly constrained). The second plot in green for Rostosky et al. reveals that this algorithm only spans snow depths from around 18 cm to 45 cm. The overall agreement is quite good, but the lack of snow depth lower than 18 cm is quite striking. The third plot in red is associated with the snow depth estimates by Kilic et al. . It reveals the widest spread of estimated snow depth values and shows a good overall agreement with the OIB distribution, but a tendency to overestimate snow depth. The plot in pink shows the snow depth distribution from our AMSR2-only neural network and the purple plot shows the distribution from a combination of the two neural networks. When SMOS data is available, we use the AMSR2 + SMOS network, otherwise just the AMSR2 network. Both neural networks show the best agreement with OIB: they capture the spread of OIB snow depths quite well, just a few snow depths deeper than 40 cm are missing and the modes are a little bit shifted. Especially a high mode at 10 cm snow depth sticks out, slightly underestimating OIB snow depth and again the combined neural network agrees a little bit better with OIB than the AMSR2-only neural network.

Finally we also plot the distribution of the deviation from OIB (estimated snow depth - OIB snow depth) in Fig. 6. The vertical black line indicates zero deviation or a perfect match between the algorithm and OIB. For clarity, we choose to use all the OIB data, since the results for the test data look similar. The neural networks show the least bias and an almost Gaussian distribution compared to OIB. Their modes are exactly at zero, while all other algorithms tend to more or less overestimate snow depth compared with the OIB measurements.

To get a better feeling for the algorithms' performance outside the areas (West Arctic) and times (spring) of the OIB data, we apply them to the whole Arctic for a whole winter season. Figure 7 shows the spatial distribution of snow depth on 11 January 2013, gridded to the 25 km polar stereographic EASE2 grid. This date was chosen arbitrarily in mid-winter. For comparison we also include the Warren climatology for January. While we do not know which solution is closest to the truth, we can see how




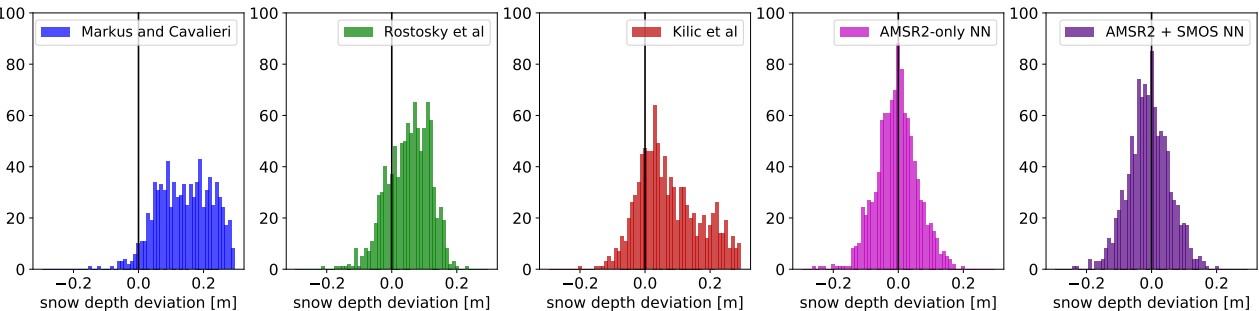

**Figure 6.** Deviation from OIB (predicted snow depth using different algorithms minus OIB measured snow depth) for all OIB data. The vertical black line indicates a perfect match between OIB and the algorithm, the left half corresponds to an underestimation and the right half to an overestimation of snow depth compared to the OIB measurements.

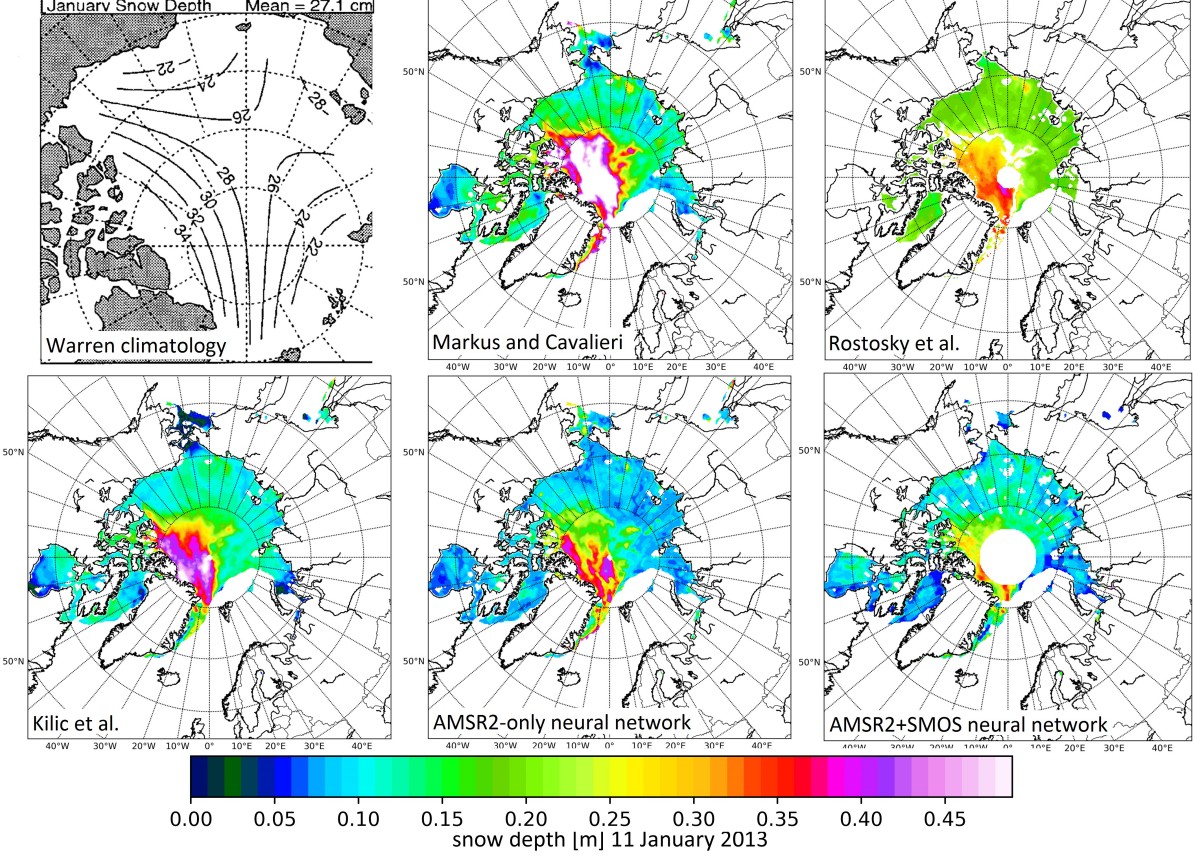

**Figure 7.** Map of Arctic snow depth on 11 January 2013 estimated with different algorithms



broad the Warren climatology is compared to the maps from satellite data. We also observe that the climatology only covers the central Arctic. Outside the diagram (e.g. 80°N on the Atlantic side) snow depth can only be calculated by extrapolation, but is no longer supported by measurements. In the plot for Markus and Cavalieri we observe a large area of snow depths exceeding 50 cm (white) in the MYI area, where this algorithm overestimates snow depth and should not be applied. We note

that Rostosky et al. lack thin snow depths of less than approximately 15 cm, which seems unrealistic in areas of young ice. The small gaps in the central Arctic and the smaller extend of the snow depth map are due to uncertainties in the ice type product. These parts are excluded in the algorithm. Rostosky et al. fill them by averaging over a month. Gaps in the AMSR2 + SMOS neural network map are due to missing SMOS data. They could be closed using the AMSR2-only neural network instead. In the case of CIMR, however, we expect the AMSR2 + SMOS neural network to produce a continuous map with no hole at the

pole which is a feature of the CIMR coverage (i.e. there will be no hole at the pole for all CIMR measurements). The snow depth maps from Kilic et al. and the neural networks look reasonable exhibiting a higher snow cover on MYI in the central Arctic and lower snow depth on FYI and in areas of new ice, as it is also recorded in the Warren climatology. Both the spatial patterns and the average snow depth of our neural networks on MYI agree well with the Warren climatology, which is based on in situ snow depth measurements. On FYI our neural networks yield lower snow depths than recorded in the climatology,

which can be explained by a strong retreat of MYI since the period when the underlying in situ data for the climatology were collected (1954-1991). Overall we can conclude that the snow depth values and the spatial pattern generated by our neural networks seem reasonable compared to both other algorithms and the Warren climatology, which is based on actual in situ measurements. However, a full validation is not possible due to a lack of ground truth data.

Figure 8 shows time series of snow depth over one winter season 2012/2013 at different locations in the Arctic. We calculate

snow depth using the different algorithms on a daily basis from the AMSR2 L1R swath data. The resulting time series have been smoothed by applying a 7-day running average to reduce noise. The first panel on the top left shows the evolution of snow depth at 65 °N, 80 °W at the entrance to Hudson Bay. As the time series reveal, a closed sea ice area started forming here only at the end of November 2012. The algorithm by Rostosky et al. gives no estimate at this position, since the ice type information is not certain enough. In general all four algorithms show an overall increase of snow depth with time, which is in

line with our expectation on FYI. The exact progression and the absolute depth vary depending on the algorithm. Most striking is, that in mid of March the neural networks and Markus and Cavalieri 's algorithm observe an increase in snow depth, while the approach by Kilic et al. leads to a decrease. Unfortunately no in situ measurements are available for comparison to verify the actual situation.

The two plots on the right show snow depth evolution on FYI at 80 °N, 80 °E in the Kara Sea (lower panel) and at 80 °N,

160 °E very close to the MYI or minimum ice extend edge of 2012 (upper panel). For all three FYI plots one might expect, that snow depth would start at zero, when the new ice has just formed. In reality most algorithms start at approximately 0.10 m and snow depth estimation from Rostosky et al. starts at 0.20 m. This can be explained by the fact that we start calculating snow depth once SIC has reached 80% and SIC algorithms are known to underestimate SIC when thin sea ice is present: Ivanova et al. (2014) showed that in case of 100% SIC, the Nasa Team algorithm will only reach 80% SIC at 0.20 m SIT, so snow depth

is not calculated before the ice has grown 20 cm thick.





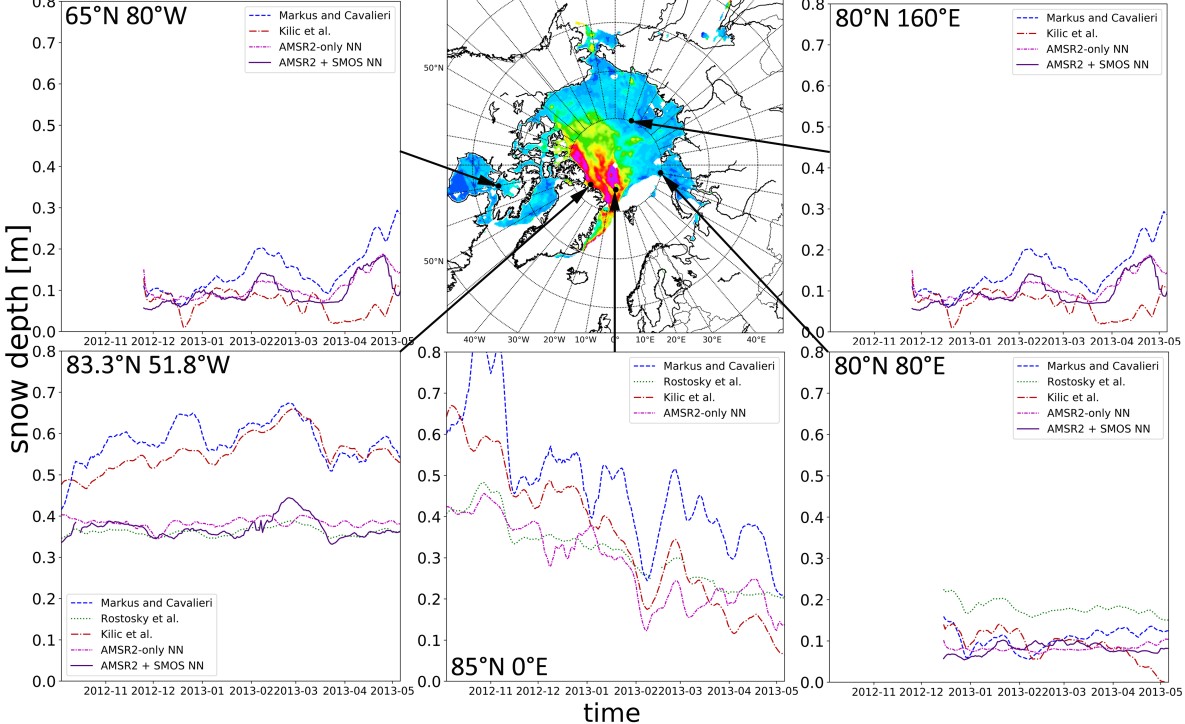

**Figure 8.** Time series of snow depth over one winter season 2012/2013 at different locations in the Arctic. The different algorithms are represented by different colours and lines. Snow depth on FYI in lower latitudes is only recorded once SIC has reached 80%.

The plots on the bottom left at 83.3 °N, 51.8 °W just North of Greenland and in the bottom center at 85 °N, 0 °E show snow depth on MYI. The left one is an example for high snow depth all year round, while the one at the center exhibits a decrease in snow depth throughout winter. This is likely because of less snow fall and a densification of the snow in winter. In general, all algorithms more or less agree on the main trends on MYI, but snow depths by Markus and Cavalieri and Kilic et al. are higher than snow depths by Rostosky et al. and our neural networks. Here again we recall that the algorithm by Markus and Cavalieri is not reliable over MYI and tends to largely overestimate snow depth larger than 50 cm.

Even though no real validation is possible over the whole Arctic or outside the OIB season, from the verification and intercomparison results we present, we can conclude that our neural network results are similar in comparison to other approaches. This indicates that, although only trained in a limited area and with spring data, the neural networks may be applied for the whole Arctic and during a full winter season.

## 4.2 Results on sea ice thickness

Having assessed the different snow depth algorithms, we now investigate how they influence SIT retrieval from CryoSat freeboard data and compare the results to OIB measured SIT. In addition to the algorithms discussed above, we also include





the Warren climatology and the OIB measured snow depth, that was used as validation snow depth data before. From the Warren climatology we use the mean monthly snow depth of 0.324 m in March and 0.337 m in April without taking any spatial variability into account. The OIB flight tracks lay within the 32 cm to 40 cm contour lines provided by Warren et al. , so using the mean value may lead to a slight underestimation of snow depth and hence also SIT, but this deviation should be within the

uncertainty range. The results are presented in Table 3 and visualised in Fig. 9.

Using the OIB measured snow depth yields the lowest bias and RMSE, the highest coefficient of determination, and the second highest correlation coefficient. Therefore using it as validation data for snow depth seems justified. However, the difference to the snow depth algorithms is not that large, when they are used in SIT retrieval. Concerning the correlation coefficient, using snow depth from Kilic et al. gives the best result, but the difference to other algorithms is marginal. In terms

of RMSE and bias, our neural networks show the second highest agreement with the OIB SIT, just after the OIB measured snow depth. For the coefficient of determination, the AMSR2-only neural network is even as good as the OIB snow product, closely followed by the AMSR2+SMOS neural network and the algorithms by Rostosky et al. and Kilic et al. . Markus and Cavalieri 's approach may perform equally well on FYI. Here we include the performance over MYI, where the algorithm is not suitable, to allow a comparison to the other approaches over all the OIB data. This explains why their approach performs

worse. For the Warren climatology we observe a large positive bias corresponding to an overestimation of SIT/snow depth. This indicates that the use of the monthly mean value has no negative impact, since we would have expected this to result in an underestimation.

When using our two neural networks' snow depth in the SIT calculation, the difference between them becomes quite small. Here the AMSR2-only neural network performs slightly better than the combined network, but not much. This is first of all

because when no SMOS data is available, both have the same result and second, in SIT calculation many other uncertainties (e.g. uncertainty in freeboard retrieval due to a varying scattering horizon within the snow pack, the choice of the retracking algorithm, uncertainty in snow density and ice density) overshadow the uncertainty in snow depth itself. In the following plots we therefore only show the AMSR2-only neural network.

In Figure 9 differences between each of the snow depth products are hard to see. Most striking is that for SITs lower than

1 m all algorithms overestimate SIT. Scatter around the measured OIB SIT is evident and the uncertainty in OIB SIT is quite large (grey shaded area).

Figure 10 shows the distribution of OIB measured snow depth in grey and the distribution of SIT calculated from CryoSat freeboard using different snow products in colour. The SIT distribution from CryoSat freeboard using the Warren climatology shows less low SIT (up to 2 m) compared to OIB measured SIT. The same holds when using snow products from Markus and

Cavalieri or Rostosky et al. , but the effect is less pronounced. Apart from that both distributions are in quite good agreement for all the snow products.

Figure 11 exhibits the deviation of the calculated SIT with CryoSat using different snow depth products from the OIB measured SIT. For most snow depth estimates, CryoSat derived SIT is higher than the OIB measured SIT. This effect can be seen clearly for the Warren climatology and the Markus and Cavalieri snow depth product, but also the algorithms by Rostosky

et al. and Kilic et al. lead to a slight overestimation of SIT. In contrast, using the OIB measured snow depth or the neural





|  | RMSE | CC | $R^2$ | bias |
|---|---|---|---|---|
| Warren climatology | 0.720 m | 0.75 | 0.51 | 0.237 m |
| Markus & Cavalieri | 0.752 m | 0.79 | 0.46 | 0.377 m |
| Rostosky et al. | 0.648 m | 0.79 | 0.62 | 0.071 m |
| Kilic et al. | 0.629 m | **0.81** | 0.62 | 0.120 m |
| *OIB snow* | *0.620 m* | *0.80* | *0.63* | *-0.054 m* |
| AMSR2-only NN | 0.625 m | 0.79 | **0.63** | -0.059 m |
| AMSR2 + SMOS NN | 0.628 m | 0.79 | 0.62 | -0.074 m |

**Table 3.** RMSE, correlation, coefficient of determination and bias between CryoSat derived SIT using different snow products and OIB measured SIT for all data.

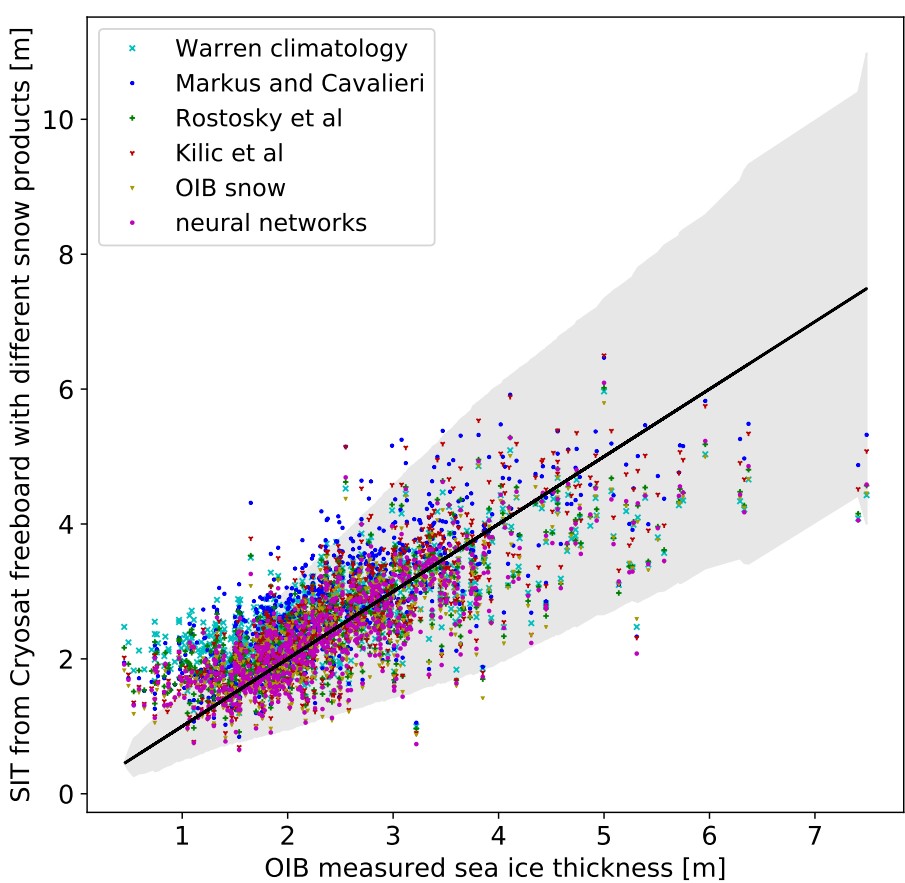

**Figure 9.** OIB measured SIT (all data) versus predicted SIT using CryoSat freeboard with different snow depth algorithms.





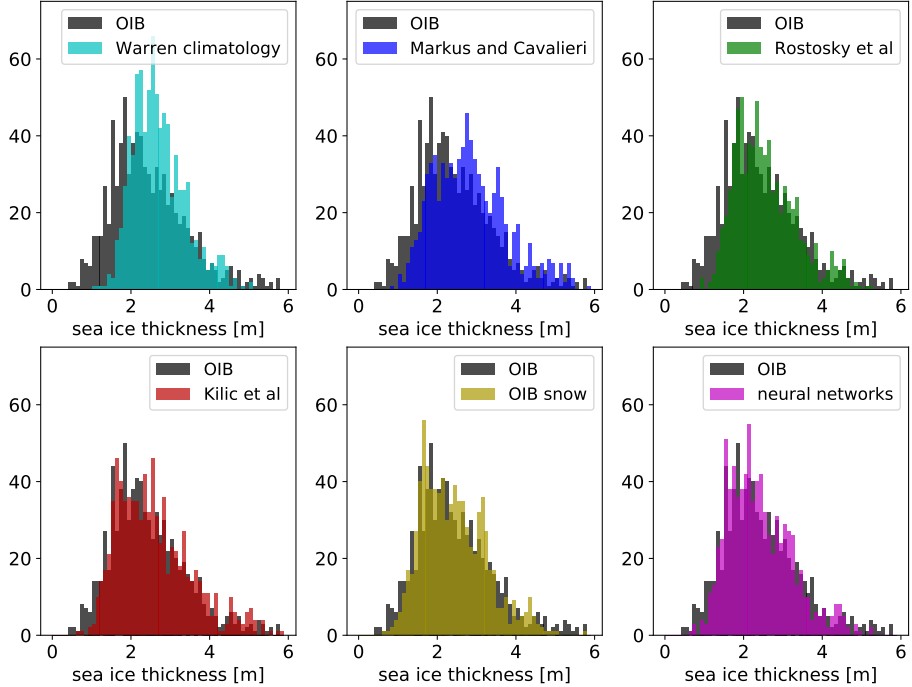

**Figure 10.** Distribution of OIB measured SIT (all data) in grey versus predicted SIT using CryoSat freeboard with different snow depth algorithms in colour.

networks in the CryoSat SIT retrieval gives almost no bias compared to OIB SIT measurements. The modes of the deviation are exactly at zero. Only a minor skew in the distribution indicates a slight underestimation of SIT.

Snow depth is not the only uncertainty in SIT estimation. A large contribution to the SIT error budget is the position of the radar scattering horizon. For CryoSat the assumption is that most of the signal is scattered at the snow-ice interface, however

different studies suggest that in some cases (e.g. with a saline snow pack, slush and layering) the main scattering horizon is rather "somewhere within the snowpack" (King et al. (2018), Price et al. (2015), Kwok and Kacimi (2018)). Alexandrov et al. (2010) state that the freeboard error may be reduced by averaging. In our comparison with OIB we take the mean of on average 296 CryoSat freeboard measurements in the collocation process, so the freeboard error should be reduced significantly, but systematic errors e.g. originating from the choice of the retracker remain (Ricker et al. (2014)). Both snow- and ice density

change spatially and temporally, but are mostly treated as a constant bulk value. This introduces further uncertainty in the conversion of CryoSat freeboard to SIT. Kern et al. (2015) found in their sensitivity study that the uncertainties of ice density and snow depth contribute about equally to SIT uncertainty. According to Zygmuntowska et al. (2014) and Giles et al. (2007) snow depth is the biggest uncertainty. With our results we also show that the snow depth product does play a role and makes a difference. This implies that using a reliable snow depth product also gives more accurate SIT. Compared with OIB we can

confirm that the neural networks give the best results both for snow depth and using this snow depth in the SIT calculation.





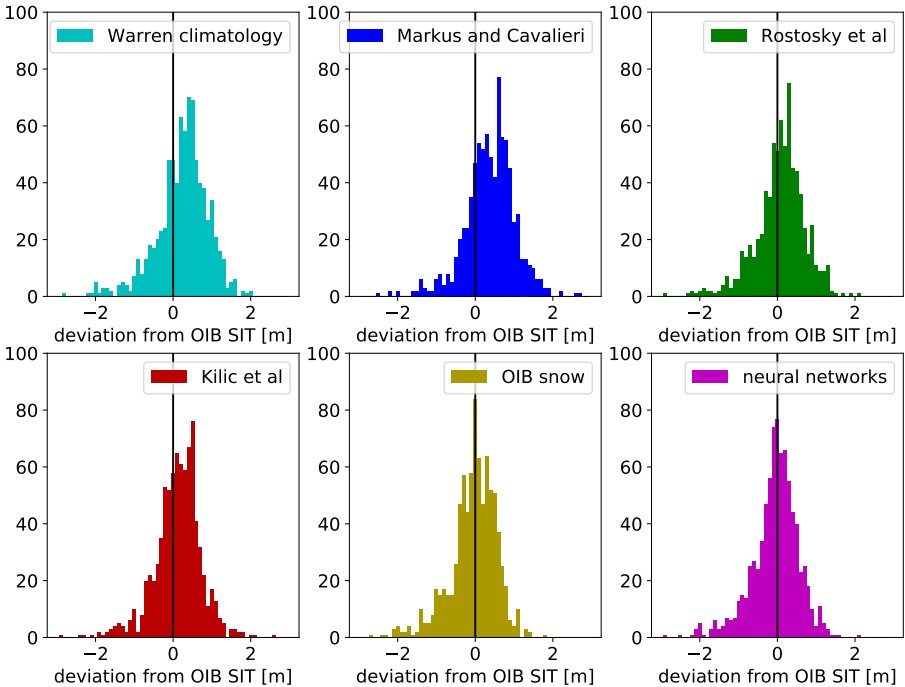

**Figure 11.** Deviation from OIB (predicted SIT using CryoSat freeboard with different snow depth algorithms minus OIB measured SIT) for all OIB data. The vertical black line indicates a perfect match between OIB and the algorithm, the left half corresponds to an underestimation and the right half to an overestimation of SIT compared to the OIB measurements.

## 5 Conclusions

In this paper we introduce a novel neural network approach to derive snow depth on sea ice from microwave radiometer brightness temperatures. We design one neural network that relies only on AMSR2 brightness temperatures and another neural network that takes brightness temperatures from SMOS as additional input. We evaluate the results with snow depth measure-
5  ments from the OIB snow radar and compare them to three other more conventional microwave radiometer algorithms.

We find that both our neural networks outperform the other algorithms when compared to OIB snow depths. The neural networks show the lowest RMSE, the highest coefficient of determination and especially have essentially no bias. The estimated snow depth covers the full range of measured OIB snow depths and our approach works over both FYI and MYI without requiring a map of ice types to distinguish between both. We also demonstrate that the neural networks are applicable outside
10  the OIB period and time, showing reasonable results, that are in line with our expectation, the other algorithms and the Warren climatology, which is based on in situ measurements, but a true validation should be subject to future work. Comparing our two neural networks with each other shows that the addition of SMOS further boosts the accuracy. The AMSR2-only neural network can be used to fill areas where no SMOS data is available.



Additionally we derive SIT from CryoSat freeboard measurements using different snow products including the algorithms tested before, the Warren climatology and our neural networks. In comparison to the SIT derived from laser altimeter measurements onboard OIB, we can confirm that using the snow depth retrieved with our neural networks also yields the best matching SIT. This underlines the importance of a reliable snow product and supports our neural network approach.

The Copernicus Imaging Microwave Radiometer (CIMR) candidate mission is now being studied at ESA. CIMR proposes a conically scanning radiometer having a swath > 1900 km and will include channels at 1.4 GHz (60 km), 6.9 and 10.65 GHz (<15 km), 18.7 GHz (5-6 km) and 36.5 GHz (4-5 km) on the same platform in a high inclination dawn-dusk orbit coordinated with the MetOp-SG(B). CIMR offers improved spatial resolution compared to AMSR2 with sub-daily coverage of the polar regions above 60°north and south. An adapted version of the AMSR2 + SMOS snow depth on sea ice neural network retrieval

would be extremely valuable – especially if used in synergy with the proposed CRISTAL dual-frequency radar altimeter dedicated to sea ice thickness retrievals. Both missions could fly in late 2020.

As future work we propose a more extensive inter-comparison of our neural network approach (and other microwave radiometry retrievals) to twin-frequency altimetry snow depth retrievals, modelling approaches and climatologies. Additionally it would be interesting to examine how the neural networks perform in Antarctica or to train a similar neural network with

Antarctic in situ data. Especially the combined AMSR2+SMOS neural network seems promising for the retrieval of deeper Antarctic snow depth on sea ice, since it incorporates low frequency channels and - in contrast to most other microwave radiometry approaches - does not exhibit a saturation of the signal at 35-50 cm snow depth.

*Code availability.*  Both the AMSR2-only and the AMSR2+SMOS neural networks are made available in h5 format on https://github.com/ AnneBF/snownet. They come with a sample Python code to read and apply them. DOI: https://zenodo.org/badge/latestdoi/170323330

*Author contributions.*  ABF developed the neural networks, implemented the comparison and validation study and wrote the paper. CD supervised her work and gave input for writing.

*Competing interests.*  The authors declare that they have no conflict of interest.

*Acknowledgements.*  This work was conducted as part of the ESA Young Graduate Trainee project "EO Study of the Polar Oceans and Sea Ice" at the European Space Agency/ESTEC, Noordwijk, the Netherlands. We thank Mark Drinkwater for his support and helpful comments

to improve the manuscript.



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
