# Peer review of "Estimating Snow Depth on Arctic Sea Ice using Satellite Microwave Radiometry and a Neural Network"

_The Cryosphere, 2019_

## Short Comment (SC1) · 10 Apr 2019

I am really happy to see work on high priority Copernicus polar mission candidates to come out - especially work pointing out synergies between different candidates. In short, the paper presents a novel way to derive the thickness of snow on sea ice – a parameter that is one of the key uncertainty contributors to sea ice thickness altimeter retrievals. Passive microwave based snow product from CIMR could complement the snow thickness estimate the dual frequency altimeter product of CRISTAL, latter being of superior resolution but worse coverage. For single frequency altimeters like Cryosat-2 and Sentinel 3 the impact of a novel PMW snow estimate, like the one presented in

this paper, would be much larger than for CRISTAL.

Whenever a new snow product emerges, it is tested against the Warren 1999 (W99) climatology, as this manuscript has done. However, I feel that there are significant shortcomings in the way W99 is handled in this paper.

Most importantly, instead of the original W99, the authors should use the modified W99 which accounts for thinner snow on FYI. All of the current CS2 SIT products use the modified W99. Reason for this is that as authors point out, original W99 has been shown to give too thick snow over the FYI areas covered by OIB by Kurtz et al. A comparison of CS-2 SIT using modified W99 and OIB SIT can be found in for example in Tilling et al 2018 ( https://doi.org/10.1016/j.asr.2017.10.051 ) where the two agree within 0.5 cm. This is in stark contrast with the 24 cm bias in table 3.

Key point of the manuscript is that the new snow product is better than the original W99. Real question is, however, if the novel snow product is better than the modified W99 currently used for the CS-2 SIT retrievals. The authors should, in my opinion, add this comparison in the next version.

Furthermore, the authors begin their SIT processing from a freeboard product in the Cryosat-2 GDR. It is reasonably hard to find the details of the processor, but the freeboard is most likely already corrected for the propagation speed of radar pulse in snow. For this, a snow estimate has been required. Authors should remove the propagation speed correction and calculate another with their own snow estimate. Or if there is no propagation speed correction in the GDR freeboard estimate, one must be applied before FB to SIT conversion.

---

## Referee Comment (RC1) · Anonymous Referee #1 · 4 May 2019

This is a well written paper that introduces a novel approach to estimating snow-on-sea-ice thickness using multi-frequency passive microwave data. The authors go on to show how better snow thickness estimates impact the further calculation of sea ice thickness. The paper is well organized, the references complete, and the figures generally clear.

The authors discuss several previous snow thickness algorithms in some detail. They compare results against OIB data to test RMSE and correlation. I think it would be useful to add additional description about the physical basis for the different algorithms. In most cases at least in so far as I recall, the algorithms are largely empirical and are

validated against in situ data. However the authors seek to use this analysis as a guide for the CIMR mission. Without more detail on the physical basis, it is hard to say how well the algorithm will perform or serve to continue as a long term record given differences between the CIMR instrument parameters and earlier sensors. See for example Zabel and Jezek, 1994, Consistency in Long Term Observations of Oceans and Ice From Space, JGR Oceans, Vol 99, p. 10109.

In several of the tables, the authors quote precision to the mm level. Given that the OIB data are at best accurate to 1 cm for snow thickness and maybe 5 cm for ice thickness, the precision in the table should be changed to reflect that. It might also be interesting to think about the accuracy of the algorithm derived snow thickness and SIT. There is uncertainty in the accuracy of the OIB data but there is also algorithmic uncertainty the arises from the assumptions in the algorithm. What might be the later and what might be the total uncertainty in the results presented here?

I recommend publication after the authors have reviewed my comments.

---

## Referee Comment (RC2) · Anonymous Referee #2 · 9 May 2019

The manuscript introduces the use of neural networks into the space of snow thickness retrieval on sea ice. The authors compare their results to previous methods including the most recent methods on this subject. In addition to the snow depth algorithm evaluations, the authors present the influence on an ice thickness estimate from CryoSat and compare it to the widely used Warren climatology. The authors also give an outlook towards the possibilities of the joint forces of the candidate missions CIMR and CRISTAL using their methods. The research is well conducted and the manuscript is well written and is suitable for publication after minor copy editing and addressing the following comments:

[Figure]

General:

1. The neural networks are trained using a small amount of data. According to the text, the data was split into train, validation and test data sets. Did you somehow ensure that that similar values of snow depth occur in all three splits? In best case the histogram of snow depth in each of the splits should be similar. Did you try different splits and compare the results?

2. It is unclear to me why the GRs and PRs are used as input. One would expect that the neural network would figure out the relations and adjusts the weights accordingly during the training process. Did you try higher complexity of the networks when you used brightness temperatures as input?

3. For comparisons with Models and also for the uncertainty values of ice thicknesses from CryoSat it is quite important to have uncertainty attached to each retrieved value. Can you think of a method estimating uncertainties for the neural network based snow thickness retrieval? It would be good to have a statement about this in the manuscript.

Specific:

P1. L.2: .....it is fundamental climate.... -> .... it is a fundamental climate....

P2. L.34: acts as -> behaves like

P11. L 23: A few words about sea ice drift as a source for ice thickness variability would be nice.

P15. L24: remove either "polar stereographic" or "EASE2". The EASE2 grid is actually not a polar stereographic projection but an equal area projection.

P16. Figure7: The AMSR2 and the AMSR2+SMOS neural networks produce very different spatial distribution of snowdepth and often by more than 20cm and even show inverse pattern (Canadian archipelago, East Greenland). To believe your statement that the combination of the two neural network would produce good results, a scatter

plot between these two networks might be insightful, especially over a longer time span.

Also P18 Figure 8 show partly anti pattern between the AMSR2 and the AMSR2+SMOS neural network. P23.

---

## Short Comment (SC2) · 31 May 2019

This paper presents and compares some very interesting and promising methods to retrieve the Snow Depth (SD) with AMSR-2. Such studies are very important because the Snow Depth over sea ice remains largely unknown whereas it plays an important role in the climate (albedo), the sea ice dynamics (thermal insulation, melt pounds), the biochemical (UV insulation), etc. But the validation of the emerging solutions is a very difficult task du to the snow diversity and the lack of in-situ data. Also we must be very careful in our conclusions and clearly stated the uncertainties and the conditions of applicability.

[Figure]

My remarks and questions are the following :

1/ Could you indicate if the presented SD-AMSR-2 products are available and where we could get them in order to make alternative tests ?

2/ Did you evaluate the product that is freely distributed by NSIDC (https://nsidc.org/data/AU_SI12/versions/1) ?

3/ How do you manage the impacts of the fog and clouds ? For instance within the NSIDC product some large parts are missing because of the presence of clouds, which is not the case for the 5 solutions you present. More generally, do they work all along the year over the full Arctic basin ?

4/ For the sea ice thickness comparisons, it seems that you are using the CryoSat-2 Baseline-C freeboard (FB), which is known to over estimate the sea ice freeboard by more than 10cm (ie, 1m on the thickness). This biais will be corrected in the next baseline-D. In the meantime, you should use other FB products (AWI, LEGOS, CPOM, NASA, JPL, ...).

5/ Due to the dramatic lack of SD data over the polar regions, all study tracks have to be investigated and the solution will most probably come from the synergy between several solutions to cover the different needs. Nevertheless, in order to improve the Sea Ice Thickness (SIT) retrieval from altimetry, it is really important to measure the SD synchronously and coherently with the FB, ie, from the same platform and the same instruments, as proposed by CRISTAL Copernicus candidate mission.

On the other hand the synergy between CRISTAL and CIMR could aim to daily pan-Arctic SD observations, which would be a major step forward to better model the dynamics of the ice pack and its snow cover, and their impact on the climate.

This kind of study could definitively participate to reach such an achievement.

---

## Author Comment (AC1) · 25 Jun 2019

**We thank the anonymous reviewer for his assessment and valuable ideas to improve and clarify the manuscript.**

*This is a well written paper that introduces a novel approach to estimating snow-on-sea-ice thickness using multi-frequency passive microwave data. The authors go on to show how better snow thickness estimates impact the further calculation of sea ice thickness. The paper is well organized, the references complete, and the figures generally clear. The authors discuss several previous snow thickness algorithms in some detail. They compare results against OIB data to test RMSE and correlation. I*

[Figure]

*think it would be useful to add additional description about the physical basis for the different algorithms. In most cases at least in so far as I recall, the algorithms are largely empirical and validated against in situ data.*

**We added more details about the physical basis of the algorithms in Section 2.1 (p.4 ll. 23-25):** "[Markus and Cavalieri (1998) developed the first algorithm to retrieve snow depth $h_s$ on sea ice from passive microwave measurements in 1998.] The physical basis of their algorithm is the fact that brightness temperature is sensitive to volume scattering. The brightness temperature over snow on sea ice decreases, when snow depth increases or when frequency decreases. They found the highest correlation to Antarctic snow depth observations with** [the gradient ratio between 19 GHz and 37 GHz brightness temperatures Tb at vertical polarization $V$:]"

**True, all algorithms presented in the paper are based on empirical fits to observation data (OIB, buoys or ship and ground observations). This is already discussed in the paper.**

*However the authors seek to use this analysis as a guide for the CIMR mission. Without more detail on the physical basis, it is hard to say how well the algorithm will perform or serve to continue as a long term record given differences between the CIMR instrument parameters and earlier sensors. See for example Zabel and Jezek, 1994, Consistency in Long Term Observations of Oceans and Ice From Space, JGR Oceans, Vol 99, p. 10109.*

**We added a more details describing the CIMR sensor in Section 3.2 (p.11 ll.7-9):** "[Since CIMR would provide the same frequencies that we are using] **(6.9, 18.7 and 36.5 GHz) at the same incidence angle ($55°$)** [and a similar L1R product, our neural networks could directly be applied to CIMR data and would provide snow depth at a higher spatial resolution]."

**Both AMSR2 and CIMR are measuring brightness temperature at $55°$ incidence**

angle and 6.9 GHz, 18.7 GHz and 36.5 GHz as used in the algorithms. The same holds for the former AMSR-E sensor. The slightly different incidence angle of SSMI ($53.1°$) and frequency (19.35 and 37.0 GHz, no C-band channel) were taken into account by using regression coefficients of the Markus and Cavalieri approach that were adapted for the AMSR instrument parameters by Comiso et al. (mentioned in Section 2.1). Both our neural networks and the algorithms by Rostosky et al. and Kilic et al. are designed for the instrument parameters of AMSR-E, AMSR2 or CIMR. To extend the long term record beyond 2001, when AMSR-E was launched, data from SMMR could be used, but the different incidence angle ($50.2°$) and slightly different frequencies (6.6, 18.0 and 37.0 GHz) might degrade their performance. Zabel and Jezek (1994) state that snow on sea ice is not too sensitive to small differences in instrument parameters, since surface roughness dominates. Evaluating this could be subject to future work.

The AMSR2+SMOS NN takes additional brightness temperature measurements at 1.4GHz from SMOS. CIMR will cover the same frequency, but at a constant incidence angle of $55°$, while SMOS is measuring at varying incidence angles between 0 and $65°$. To simulate brightness temperatures as they will be measures by CIMR we only took measurements from SMOS between 50 and $60°$ incidence angle and averaged them. This is described in Section 3.3. Therefore we are confident that the instrument parameters of AMSR2 and SMOS are close enough to CIMR.

*In several of the tables, the authors quote precision to the mm level. Given that the OIB data are at best accurate to 1 cm for snow thickness and maybe 5 cm for ice thickness, the precision in the table should be changed to reflect that.*

**The precision in the tables and the text has been changed. For the SIT we kept cm precision, but point to the lower accuracy of the OIB data in the text:** "[In terms of RMSE our AMSR2-only neural network performs as good as the OIB snow product and both the algorithm by Rostosky et al. and the AMSR2+SMOS neural network are

only 1 cm worse,] **which is not significant considering that the accuracy of the OIB SIT is at best 5 cm. Therefore the last digit of the bias and the RMSE should not be overrated."** **And** "[When using our two neural networks' snow depth in the SIT calculation, the difference between them becomes marginal] **and is smaller than the accuracy of the OIB SIT."**

*It might also be interesting to think about the accuracy of the algorithm derived snow thickness and SIT. There is uncertainty in the accuracy of the OIB data but there is also algorithmic uncertainty the arises from the assumptions in the algorithm. What might be the later and what might be the total uncertainty in the results presented here? I recommend publication after the authors have reviewed my comments.*

**We divided the former chapter 4.1. into two subsections and added a third subsection on uncertainty estimation at the end of chapter 4.1 (as 4.1.3.) (p.18, l.10): "Finally we assess the uncertainty of our neural networks to enable usage of this snow product in models or for SIT calculation. The very complex and highly nonlinear relationship between the input and snow depth output hinders a stringent variance propagation. Instead, to assess the uncertainty of our neural network approaches, we employ the Monte Carlo method and generate an ensemble of 50 samples for each input brightness temperature. We draw these samples from a normal distribution using the observed brightness temperature as mean and 0.5 K as standard deviation for AMSR2.**

**For SMOS we take the standard deviation provided in the L3 files for each observation and propagate them through our averaging process to obtain a standard deviation for each SMOS measurement used as input to the polarization ratio. The mean of these standard deviations is 0.76 K for V polarization and 0.79 K for H polarization. Further uncertainty arises from the tie points used both in the Nasa Team algorithm for SIC and to correct the open water part of the footprint (Eq. 2). Therefore we also create an ensemble of 50 samples for each tie point using the values from Ivanova et al. as mean and 3 K as standard deviation. The**

resulting mean uncertainty in SIC from the Nasa Team algorithm is 4 %.

We then estimate snow depth using each ensemble member as input to our neural networks. This yields an ensemble of snow depth estimates. The standard deviation of this ensemble is used as an uncertainty measure for the estimated snow depth value. Across all OIB data the resulting final uncertainty (mean standard deviation) is 0.05 m for the AMSR2-only NN and 0.02 m for the AMSR2+SMOS NN, indicating that the AMSR2+SMOS NN is less sensitive to noise in the input data. The error (repeatability) of the Monte Carlo simulation is 0.0005 m and 0.0001 m respectively. This approach however only assesses how robust the neural networks are to uncertainty in the input data and auxiliary parameters such as the tie points. Further uncertainty arises from training the neural networks with OIB data, which has its own uncertainty and limitations unlike a real ground truth dataset. "

In the conclusion we added a sentence (p.22, l.11): "From a Monte Carlo simulation we derive an uncertainty of 5 cm for the AMSR2-only and 2 cm for the AMSR2+SMOS NN."

We also updated the sample code provided on github (mentioned under code availability) to include a standard deviation for each snow depth estimate (p.23, ll.18-19).

---

## Author Comment (AC2) · 25 Jun 2019

**We thank the anonymous reviewer for his insightful comments and ideas to improve the manuscript.**

*The manuscript introduces the use of neural networks into the space of snow thickness retrieval on sea ice. The authors compare their results to previous methods including the most recent methods on this subject. In addition to the snow depth algorithm evaluations, the authors present the influence on an ice thickness estimate from CryoSat and compare it to the widely used Warren climatology. The authors also give an outlook towards the possibilities of the joint forces of the candidate missions CIMR and*

*CRISTAL using their methods. The research is well conducted and the manuscript is well written and is suitable for publication after minor copy editing and addressing the following comments:*

*General: 1. The neural networks are trained using a small amount of data. According to the text, the data was split into train, validation and test data sets. Did you somehow ensure that that similar values of snow depth occur in all three splits? In best case the histogram of snow depth in each of the splits should be similar. Did you try different splits and compare the results?*

**We had a look at the histograms of different splits and added a few comments on this in section 3.1 (p.10, ll.10-15, p.11 l.1):**

"[To train the networks we temporally divide the OIB data into a training- (70%), a validation- (15%) and a test (15%) dataset.] **This is a common splitting in machine learning and ensures enough training data when the overall amount of data is small. We also verified that each of the splits contains a similar range of snow depth values and that their histograms look alike.** [Figure 3 shows the flight tracks and the measured snow depths from the 2013-2015 campaigns] **(the overall dataset).** [The top right box illustrates which parts are used as test data] **and the snow depth values occurring in this split.** [We end up with 755 valid snow depth measurements in 2013 and 2014 for training, 162 valid measurements in 2014 and 2015 for validation – meaning the identification of the best network architecture – and 162 valid snow depth measurements in 2015 for testing. When we train the AMSR2 + SMOS neural network, we have to discard all areas (especially the bigger hole at the pole), where no SMOS data is available. Again we split the remaining data into 70% for training and 15% for validation and testing each] **and confirm similar histograms of all splits."**

**We found that a 60-20-20 split looks similar, but would give less training data and an 80-10-10 split for example yields histograms that are less alike. Attached (Figure 1) you can find the histograms of our 70-15-15 split. For the AMSR2+SMOS**

NN we have less data available due to SMOS's bigger hole at the pole. However, a 70-15-15 split of the remaining data also gives histograms very similar to the ones shown. For the paper, we believe that figure 3 is sufficient to show which values occur in the overall and in the test data set.

*2. It is unclear to me why the GRs and PRs are used as input. One would expect that the neural network would figure out the relations and adjusts the weights accordingly during the training process. Did you try higher complexity of the networks when you used brightness temperatures as input?*

We also tried more complex neural networks when using brightness temperatures as input, but they have reduced performance. Our explanation for this is that a more complex neural network also implies that more parameters have to be learned with the same (very limited) amount of training data.

*3. For comparisons with Models and also for the uncertainty values of ice thicknesses from CryoSat it is quite important to have uncertainty attached to each retrieved value. Can you think of a method estimating uncertainties for the neural network based snow thickness retrieval? It would be good to have a statement about this in the manuscript.*

We divided the former chapter 4.1. into two subsections and added a third subsection on uncertainty estimation at the end of chapter 4.1 (p.18, l.10): "Finally we assess the uncertainty of our neural networks to enable usage of this snow product in models or for SIT calculation. The very complex and highly non-linear relationship between the input and snow depth output hinders a stringent variance propagation. Instead, to assess the uncertainty of our neural network approaches, we employ the Monte Carlo method and generate an ensemble of 50 samples for each input brightness temperature. We draw these samples from a normal distribution using the observed brightness temperature as mean and 0.5 K as standard deviation for AMSR2.

For SMOS we take the standard deviation provided in the L3 files for each observation and propagate them through our averaging process to obtain a standard deviation for each SMOS measurement used as input to the polarization ratio. The mean of these standard deviations is 0.76 K for V polarization and 0.79 K for H polarization. Further uncertainty arises from the tie points used both in the Nasa Team algorithm for SIC and to correct the open water part of the footprint (Eq. 2). Therefore we also create an ensemble of 50 samples for each tie point using the values from Ivanova et al. as mean and 3 K as standard deviation. The resulting mean uncertainty in SIC from the Nasa Team algorithm is 4 %.

We then estimate snow depth using each ensemble member as input to our neural networks. This yields an ensemble of snow depth estimates. The standard deviation of this ensemble is used as an uncertainty measure for the estimated snow depth value. Across all OIB data the resulting final uncertainty (mean standard deviation) is 0.05 m for the AMSR2-only NN and 0.02 m for the AMSR2+SMOS NN, indicating that the AMSR2+SMOS NN is less sensitive to noise in the input data. The error (repeatability) of the Monte Carlo simulation is 0.0005 m and 0.0001 m respectively. This approach however only assesses how robust the neural networks are to uncertainty in the input data and auxiliary parameters such as the tie points. Further uncertainty arises from training the neural networks with OIB data, which has its own uncertainty and limitations unlike a real ground truth dataset. "

In the conclusion we added a sentence (p.22, l.11): "From a Monte Carlo simulation we derive an uncertainty of 5 cm for the AMSR2-only and 2 cm for the AMSR2+SMOS NN."

We also updated the sample code provided on github (mentioned under code availability) to include a standard deviation for each snow depth estimate (p.23, ll.18-19).

*Specific: P1. L.2: .....it is fundamental climate.... -> .... it is a fundamental climate....*

**Changed**

*P2. L.34: acts as -> behaves like*

**Changed**

*P11. L 23: A few words about sea ice drift as a source for ice thickness variability would be nice.*

**We added the following sentence:**

**"In areas of mixed ice types and fast sea ice drift this assumption might not hold, but we want to avoid too many data gaps."**

*P15. L24: remove either "polar stereographic" or "EASE2". The EASE2 grid is actually not a polar stereographic projection but an equal area projection.*

**Polar stereographic was removed**

*P16. Figure7: The AMSR2 and the AMSR2+SMOS neural networks produce very different spatial distribution of snow depth and often by more than 20cm and even show inverse pattern (Canadian archipelago, East Greenland). To believe your statement that the combination of the two neural network would produce good results, a scatter plot between these two networks might be insightful, especially over a longer time span. Also P18 Figure 8 show partly anti pattern between the AMSR2 and theAMSR2+SMOS neural network. P23.*

**We have produced a scatter plot between the AMSR2-only and AMSR2+SMOS neural networks on all the OIB data (Figure 2). For comparison we also plot OIB snow depth against the AMSR2-only NN and find that both scatter to a similar extend around the AMSR2-only NN results. Indeed, a few estimates are quite far apart, but the same is true for the OIB data. Given the results presented in the paper (Table 2, Figure 5 and 6) we believe, that filling gaps in the AMSR2+SMOS NN with the AMSR2-only NN produces reasonable results. One main purpose**

**of the AMSR2+SMOS NN development, however, was to investigate how snow depth on sea ice could be derived from CIMR measurements and here we will not encounter the problem of 1.4GHz data gaps.**

**We added another sentence in Section 4.1.(1) (p.13, ll. 10-11) to stress this point:** "[Combining the two networks could also be useful in a practical application to fill the hole at the pole and to still benefit from higher accuracies in regions, where SMOS is available.] **CIMR, however, would cover the whole pole at all frequencies and therefore, the AMSR2 + SMOS neural network would produce no gaps.** "

————————————————————

[Figure]

**Fig. 1.** Histograms of 70% training, 15% validation and 15% test data splits

[Figure]

**Fig. 2.** Scatter plot between AMSR2-only and AMSR+SMOS NN snow depths with OIB snow depth for comparison

---

## Author Comment (AC3) · 25 Jun 2019

**We would like to thank Eero Rinne for his insightful comments and for opening the discussion.**

*I am really happy to see work on high priority Copernicus polar mission candidates to come out - especially work pointing out synergies between different candidates. In short, the paper presents a novel way to derive the thickness of snow on sea ice – a parameter that is one of the key uncertainty contributors to sea ice thickness altimeter retrievals. Passive microwave based snow product from CIMR could complement the snow thickness estimate the dual frequency altimeter product of CRISTAL, latter being*

*of superior resolution but worse coverage. For single frequency altimeters like Cryosat-2 and Sentinel 3 the impact of a novel PMW snow estimate, like the one presented in this paper, would be much larger than for CRISTAL. Whenever a new snow product emerges, it is tested against the Warren 1999 (W99) climatology, as this manuscript has done. However, I feel that there are significant shortcomings in the way W99 is handled in this paper. Most importantly, instead of the original W99, the authors should use the modified W99which accounts for thinner snow on FYI. All of the current CS2 SIT products use the modified W99. Reason for this is that as authors point out, original W99 has been shown to give too thick snow over the FYI areas covered by OIB by Kurtz et al. A comparison of CS-2 SIT using modified W99 and OIB SIT can be found in for example in Tilling et al 2018 ( https://doi.org/10.1016/j.asr.2017.10.051 ) where the two agree within 0.5 cm. This is in stark contrast with the 24 cm bias in table 3.Key point of the manuscript is that the new snow product is better than the originalW99. Real question is, however, if the novel snow product is better than the modifiedW99 currently used for the CS-2 SIT retrievals. The authors should, in my opinion, add this comparison in the next version.*

**In the revised version of the paper, we are using the modified W99 climatology in addition to the original one. This indeed leads to an improved agreement with the OIB data, but does not change our conclusions. Our RMSE between the modified W99 and OIB are in agreement with the one reported by Tiling et al. 2018 (0.66 m and 0.67 m). Our bias (0.16 m) is indeed higher, but we do not expect to reproduce their numbers, since a few processing steps differ and we compare results for 2013-2015, while they compare their estimates with OIB data from 2011-2013. Apart from a very good agreement with OIB, they also retrieve a bias of 0.21 m compared to CryoVex, so we believe our results are plausible.**

**Adding a comparison to the modified Warren climatology leads to the following additions in the paper: In section 2.5 (p.9, ll. 17-18):** "[The last - and a major - uncertainty in the calculation of SIT is snow depth $h_s$ (Zygmuntowska et al. (2014),

Giles et al. (2007)). Here we use the] **original Warren climatology, its modified version where snow depth is halved over FYI and the algorithms from [. . .].''**

**In section 3.5.2 (p.12, l.7): "The same product is also used to modify the Warren climatology. In areas of FYI we half the original snow depth values."**

**In section 4.2 another line was added to the table and an additional subplot was added to Figure 9 and 10 to include the modified Warren climatology. In the text we added the following part (p.19, l. 15): "For the Warren climatology we observe that the modified version performs better in all the categories, but still worse than most other algorithms."**

**Also a citation was added (p.25, l.22): "Kurtz, N. T. and Farrell, S. L.: Large-scale surveys of snow depth on Arctic sea ice from operation IceBridge, Geophysical Research Letters, 38, https://doi.org/10.1029/2011GL049216, (2011)"**

*Furthermore, the authors begin their SIT processing from a freeboard product in the Cryosat-2 GDR. It is reasonably hard to find the details of the processor, but the freeboard is most likely already corrected for the propagation speed of radar pulse in snow. For this, a snow estimate has been required. Authors should remove the propagation speed correction and calculate another with their own snow estimate. Or if there is no propagation speed correction in the GDR freeboard estimate, one must be applied before FB to SIT conversion.*

**This is a valid point and we put a considerable amount of effort into finding out the details of the GDR processing. The official statement from EOhelp on the propagation speed correction is, that they don't do any. Adding a snow propagation speed correction to the freeboard data, however, results in a considerable bias independent of the snow product used. In a paper by Kwok (2014), the effect of both such a snow delay correction and snow penetration correction are discussed.**

We included these findings in chapter 2.5 (p.9, l.20): "For the calculation of sea ice freeboard $h_{fb}$ from radar freeboard $h_{rfb}$ two corrections should be applied (Kwok (2014)). The first correction $dh_p$ accounts for penetration issues caused by the scattering of the Ku-band radar signal at the air-snow interface and within the snow layer. This shifts the retracking point closer to the satellite. The second correction $dh_d$ adjusts the radar freeboard for the slower propagation speed of the radar signal within a snow layer:

$$h_{fb} = h_{rfb} + dh_p + dh_d$$

Both corrections have opposite signs and therefore more or less cancel out depending on the snow depth, the retracker and the ratio between the snow-ice and snow-air interface peaks (Kwok (2014)). It is especially hard to apply the first correction, since the ratio between the snow-ice and snow-air interface peaks is not known. Kwok's simulations suggest that for snow depths of 5-30 cm (which covers a major part of the OIB data) both corrections add up to 0.2 cm on average and are almost independent of snow depth, when a leading edge retracker is used. Therefore we apply a joint correction of 0.2 cm to all CryoSat radar freeboard data."

The uncertainty arising from this mean correction is again mentioned in the discussion in 4.2 (p.21, l.9): "Additionally we only apply a mean correction for the combined effect of radar penetration and radar delay caused by the snow pack. The sign and magnitude of this combined correction, however, depend on the snow depth and primarily the ratio between the snow-ice and snow-air interface peaks. The lack of data for the latter add to the uncertainty budget of SIT."

The citation was added as well (p.25, l.25): "Kwok, R.: Simulated effects of a snow layer on retrieval of CryoSat-2 sea ice freeboard, Geophysical Research Letters, doi:10.1002/2014GL060993 (2014)"

---

## Author Comment (AC4) · 25 Jun 2019

**We thank Sara Fleury for her contribution to the discussion, adding further suggestions and arguments.**

*This paper presents and compares some very interesting and promising methods to retrieve the Snow Depth (SD) with AMSR-2. Such studies are very important because the Snow Depth over sea ice remains largely unknown whereas it plays an important role in the climate (albedo), the sea ice dynamics (thermal insulation, melt pounds), the biochemical (UV insulation), etc. But the validation of the emerging solutions is a very difficult task du to the snow diversity and the lack of in-situ data. Also we must be*

[Figure]

*very careful in our conclusions and clearly stated the uncertainties and the conditions of applicability.*

*My remarks and questions are the following :*

*1/ Could you indicate if the presented SD-AMSR-2 products are available and where we could get them in order to make alternative tests ?*

**The neural networks have been uploaded and are publicly available together with sample Python code to read and apply them as mentioned under code availability.**

*2/ Did you evaluate the product that is freely distributed by NSIDC (https://nsidc.org/data/AU_SI12/versions/1) ?*

**We did not explicitly evaluate this product, because it is based on the algorithm by Markus and Cavalieri. We would expect the same result as shown for Markus and Cavalieri with the difference that in the NSIDC product snow depth over MYI are filtered out. We did not filter these parts in our paper to compare all algorithms over the same (and a larger) dataset.**

*3/ How do you manage the impacts of the fog and clouds ? For instance within the NSIDC product some large parts are missing because of the presence of clouds, which is not the case for the 5 solutions you present. More generally, do they work all along the year over the full Arctic basin ?*

**All snow depth on sea ice algorithms relying on passive microwave measurements are restricted to dry snow conditions and therefore the winter season (approx. mid Oct to mid May). Clouds and fog should not pose a problem, though, especially when lower frequencies are used. The large gaps in the NSIDC product are due to the fact that the Markus and Cavalieri approach does not work well over MYI (as mentioned above). Given the fact, that both our neural networks and the algorithm from Rostosky et al. are trained with OIB data, which are only**

[Figure]

**available for the Western Arctic, we cannot guarantee that the same functional relationships hold for the full Arctic basin. To see how the different algorithms behave outside the training area and period, we applied them for a whole season and across the whole Arctic (pages 15-18). We cannot really evaluate if the results are correct due to a lack of in situ data, but we do observe a generally good consistency between the algorithms and to the Warren climatology.**

*4/ For the sea ice thickness comparisons, it seems that you are using the CryoSat-2 Baseline-C freeboard (FB), which is known to over estimate the sea ice freeboard by more than 10cm (ie, 1m on the thickness). This bias will be corrected in the next baseline-D. In the meantime, you should use other FB products (AWI, LEGOS, CPOM,NASA, JPL, ...).*

**The SIT part is meant to be only a minor part in the paper and may also be understood as an outlook. You are more than welcome to use our neural networks, which are publicly available, to investigate their use with other freeboard data. We believe, that in the spirit of science an independent evaluation would be best anyway.**

*5/ Due to the dramatic lack of SD data over the polar regions, all study tracks have to be investigated and the solution will most probably come from the synergy between several solutions to cover the different needs. Nevertheless, in order to improve the Sea Ice Thickness (SIT) retrieval from altimetry, it is really important to measure the SD synchronously and coherently with the FB, ie, from the same platform and the same instruments, as proposed by CRISTAL Copernicus candidate mission.*

**We agree on that. However, sea ice thickness is not the only application where snow depth estimates are needed. Models and Forecasts for example might need (sub-) daily maps covering the whole Arctic, which CIMR would offer. Furthermore both satellites should be used in synergy and for inter-calibration of the snow depth products.**

[Figure]

*On the other hand the synergy between CRISTAL and CIMR could aim to daily pan-Arctic SD observations, which would be a major step forward to better model the dynamics of the ice pack and its snow cover, and their impact on the climate. This kind of study could definitively participate to reach such an achievement.*

---

## Author Response (AR2)

**Response to the Editor Lars Kaleschke:**

**We thank the editor Lars Kaleschke for his careful reading and his comments to further improve the paper.**

*Comments to the Author:*
*Dear colleagues,*

*thank you for carefully considering the comments and questions of the referees. I have some additional comments and suggestions for a minor revision.*

*The neural network method works well if the training data are from similar conditions. Therefore, the method works well for the springtime central Arctic where aircraft data are available. However, it is questionable if this is the case for different seasons and regions (i.e. for ice formed from low salinity water on the Siberian shelf). For SMOS we have investigated the sensitivity of brightness temperature to several parameters (see Maaß et al. (2013) and her PhD thesis). Knowing the snow (or ice) surface temperature is of great importance for the snow thickness retrieval from 1.4 GHz. This information does not go (directly) into your neural network and therefore causes uncertainties. I therefore doubt that the current neural network approach is the best choice for CIMR and instead suggest to ask for further research on methods. You may write sentences like the following in a more neutral way and consider physical-based radiative transfer techniques as a perhaps/likely superior choice for the retrieval of sea ice parameters from CIMR.*
*"An adapted version of the AMSR2 + SMOS snow depth on sea ice neural network retrieval would be extremely valuable"*

**Indeed, the snow (or ice) surface temperature, is no direct input to our neural networks, however, the neural networks might derive it internally. Comiso et al. (2003) for example derive the ice surface temperature from brightness temperature at 6GHz vertical polarization and sea ice concentration and Kilic et al. (2018b) derive the snow-ice interface temperature from brightness temperature at 18GHz vertical polarization and snow depth. The neural network has this information, too and could use it internally.**

**The comparison to other state-of-the-art algorithms from the literature suggest, that our neural networks would be the best choice for CIMR at the moment. Nevertheless we agree, that it would be interesting to see if this performance can be further improved by future research using e.g. radiative transfer techniques and added the following sentence to the outlook:**

**"Also the exploration of radiative transfer techniques using a combination of several channels could be subject to future work."**

*Table 1+2): What is the meaning of negative R²? Please check!*

**With the definition of $R^2$ as defined in the paper, negative values may occur and mean that the model fits the data worse than the mean value would. This is in line with figure 4.**

*Tables should have headings and not captions.*

**Has been changed.**

*References are not complete, all refs should include DOIs where available.*

**DOIs were added wherever available.**

*Incorrect use of \citet, please check.*

**Has been changed.**

*Best regards*

*Lars Kaleschke*

[revised manuscript text omitted]